# Revolutionizing Tomato Cultivation: CRISPR/Cas9 Mediated Biotic Stress Resistance

**DOI:** 10.3390/plants13162269

**Published:** 2024-08-15

**Authors:** Abdelrahman Shawky, Abdulrahman Hatawsh, Nabil Al-Saadi, Raed Farzan, Nour Eltawy, Mariz Francis, Sara Abousamra, Yomna Y. Ismail, Kotb Attia, Abdulaziz S. Fakhouri, Mohamed Abdelrahman

**Affiliations:** 1Biotechnology School, Nile University, 26th of July Corridor, Sheikh Zayed City 12588, Giza, Egypt; a.wael2129@nu.edu.eg (A.S.); a.farag2152@nu.edu.eg (A.H.); n.hazzanagi2161@nu.edu.eg (N.A.-S.); n.khalil@nu.edu.eg (N.E.); m.francis@nu.edu.eg (M.F.); s.yasser2103@nu.edu.eg (S.A.); yyoussef@nu.edu.eg (Y.Y.I.); 2Department of Clinical Laboratory Sciences, College of Applied Medical Sciences, King Saud University, Riyadh 11433, Saudi Arabia; 3Center of Excellence in Biotechnology Research, King Saud University, Riyadh 11451, Saudi Arabia; kattia@ksu.edu.sa (K.A.); afakhouri@ksu.edu.sa (A.S.F.); 4Department of Biomedical Technology, College of Applied Medical Sciences, King Saud University, Riyadh 12372, Saudi Arabia

**Keywords:** CRISPR/Cas9, biotic stress, genome editing, *Solanum lycopersicon esculentum* L.

## Abstract

Tomato (*Solanum lycopersicon* L.) is one of the most widely consumed and produced vegetable crops worldwide. It offers numerous health benefits due to its rich content of many therapeutic elements such as vitamins, carotenoids, and phenolic compounds. Biotic stressors such as bacteria, viruses, fungi, nematodes, and insects cause severe yield losses as well as decreasing fruit quality. Conventional breeding strategies have succeeded in developing resistant genotypes, but these approaches require significant time and effort. The advent of state-of-the-art genome editing technologies, particularly CRISPR/Cas9, provides a rapid and straightforward method for developing high-quality biotic stress-resistant tomato lines. The advantage of genome editing over other approaches is the ability to make precise, minute adjustments without leaving foreign DNA inside the transformed plant. The tomato genome has been precisely modified via CRISPR/Cas9 to induce resistance genes or knock out susceptibility genes, resulting in lines resistant to common bacterial, fungal, and viral diseases. This review provides the recent advances and application of CRISPR/Cas9 in developing tomato lines with resistance to biotic stress.

## 1. Introduction

Tomato (*Solanum lycopersicon* L.) is one of the most important vegetable crops, with an annual production of 182.3 million tons worldwide [1,2]. Globally, tomato ranks as the second most consumed vegetable, with high consumption in some countries, including China, India, North Africa, the Middle East, the US, and Brazil [1,2]. The increasing demand for tomatoes is driven by the fruit’s attractiveness and quality [3]. Unfortunately, this rising demand must be met with the current limited land and water resources.

Living organisms, particularly viruses, bacteria, fungi, nematodes, insects, arachnids, and weeds, are the main sources of biotic stress in plants. These biotic stress agents directly deprive their host of its nutrients, leading to reduced plant vigor and, in extreme cases, the death of the host plant [4,5,6]. Biotic stress not only significantly reduces crop quality and yield but also changes plant physiology, decreases biomass, reduces seed set, and causes the accumulation of protective metabolites [7,8]. According to research released by the Food and Agriculture Organization (FAO), biotic stress results in a 20% to 40% loss of the world’s crop production each year. Plants respond to biotic stress with a systemic reaction that includes the generation of reactive oxygen species (ROS), increased cell lignification to restrict pathogen transmission, and a reduction in host sensitivity. Despite these strategies, pathogens can modify the signaling pathways of plants and overcome the plants’ immune response [9]. Developing biotic stress-tolerant plants via genome editing could be achieved by the knockout of susceptibility genes [10,11], knocking-in resistance genes [12], disruption of pathogenicity genes, and/or accelerating effector-triggered immunity [13].

Over the last century, several breeding programs have been employed to enhance tomato productivity to meet the substantial increase in demand. These programs include chemical or irradiation mutagenesis, translocation breeding, and intergeneric crosses [14], which rely on mutant and/or natural-induced genetic variations to select favorable genetic combinations. These methods had various limitations; for example, large portions of the genome are occasionally transferred instead of a single gene, and thousands of nucleotides are sometimes modified instead of a single nucleotide [15]. On account of these limitations, a more advantageous method like transformation was extensively used as it is comparatively cheap, easy, and produces low copies and precise DNA integrations [16]. Nevertheless, transformation still has severe limitations, including its low rate of homologous recombination, insertion of large foreign Ti plasmid size, and low copy number [16,17]. Sequence-specific nuclei (SSN) genome editing tools have overcome these limitations, making them the most widely used tools for genome editing in plants [18].

SSNs, including meganucleases, zinc finger nucleases (ZFNs), transcription activator-like effector nucleases (TALENs), and clustered regularly interspaced palindromic repeat (CRISPR)/-associated (Cas) protein systems have become widely exploited in plant research. Applications of genome editing have significantly facilitated gene function characterization and precision breeding [19,20]. Notably, the re-evaluation of the genes, which are essential for the domestication and crop improvement of cultivated tomatoes and their wild relatives using CRISPR/Cas9, is a breakthrough in plant breeding [21,22,23]. Different genome editing tools, including ZFN, TALEN, CRISPR/Cas, and cytidine base editor, have been exploited in the research and inbreeding of *S. lycopersicum* [24].

Several publications have reported the advancement of different approaches to developing biotic stress-resistant tomato plants. Amoroso et al. focused on the advancements in genomic approaches such as molecular mapping, marker and genomic-aided breeding, and bioinformatics [25]. Since the first publications of successful utilization of genome editing tools in plants, research has been oriented on its application for crop improvement against biotic stresses. Several publications have reviewed genome editing in advancing tomato traits [26,27,28,29]; however, this review article focuses specifically on biotic stress-edited genes. In this review, we highlight recent advances and progress in CRISPR/Cas9 as a versatile genome editing tool. We further provide recent applications of CRISPR/Cas9 to enhance different biotic stress resistance in tomatoes.

## 2. Mechanism and Mode of Action of CRISPR/Cas9 Technology

The CRISPR/Cas9 system, the most effective gene-editing tool, is also known as clustered regularly interspaced short palindromic repeat (CRISPR). It is a type II CRISPR system. CRISPR/Cas9 is the first characterized system that exploits a single protein as the Cas effector. It is a powerful and efficient defense system in bacteria and archaea as adaptive immunity [30]. Naturally, when the bacteria are infected by an exogenous phage, the system is triggered by processing a single guide RNA (sgRNA) molecule that targets specific sequences in the DNA and initiates the transcription of the Cas9 nuclease. The sgRNA guides Cas9 to create double-strand breaks (DSBs) at the recognition site where the protospacer-adjacent motifs (PAMs) are present in the target DNA and then triggers degradation of the DNA sequence of the invading phages [31]. Hryhorowicz et al. provide more details regarding the CRISPR/Cas9 immune system in bacteria [32].

In 2012, Gasiunas et al. [33] and Jinek et al. [34] purified Cas9, derived from *Streptococcus thermophilus* or *Streptococcus pyogenes*, which were guided by sgRNAs to cleave target DNA in vitro. Cas9 requires a PAM sequence. For CRISPR/Cas9 to function as a genome editing tool, it requires both the PAM sequence and RNA-DNA complementary base pairing with a 20-base pairing between the sgRNA and the complementary target DNA [34]. The PAM sequence is a short nucleotide sequence (3 bases) positioned at the 3′ end of the target sequence that varies for each CRISPR/Cas9 system in order for them to work as genome editing tools [35]. The sgRNA is assembled by joining crRNA and the tracrRNA with the artificial tetraloop (lower-case nucleotides) and is also constructed to be complementary to the desired DNA sequence, directing the Cas9 nuclease to a selected editing site, as shown in Figure 1. Cas9 nuclease cleaves the DNA at its targeted location like molecular scissors generating DSB [34,35,36,37]. The DSBs activate the repair machinery of the cell’s DNA that could be used to make specific modifications to the genome. Subsequently, the double-strand breaks in the target DNA are repaired via one of two pathways: homology-directed repair (HDR), which allows for precise modifications to a plant’s genome, or non-homologous end joining (NHEJ), which results in gene disruptions or insertions/deletions (indels). Thus, the CRISPR genome editing technique consists of three major steps: sgRNA design and assembly, delivery of the CRISPR components, and target DNA editing.

The CRISPR/Cas9 system has become a routinely used system applied for gene editing in various genotypes. The system starts with designing the sgRNA that is complementary to the target segment, taking into consideration minimizing the off-targets and secondary structure possibilities. Several online bioinformatics tools are available to be utilized to select the most optimal sequence, such as CCTop (https://cctop.cos.uni-heidelberg.de:8043/, accessed 20 July 2024) [38] and CHOPCHOP (https://chopchop.cbu.uib.no/, accessed 20 July 2024) [39]. Two sgRNAs might be generated for the same target for more precise knockout and/or for the insertion of a new segment for gain-of-function. Golden Gate cloning [19,40] or Gibson assembly [41,42] techniques might be used for construct assembly.

Several methods are used to deliver CRISPR/Cas9 into tomato plants. Constructs are delivered by an *Agrobacterium*-mediated transformation method into tomato cotyledons [43]. In another study, a transient transformation assay was performed on the explants obtained from the cotyledons of tomato, using the particle gun bombardment method to deliver the CRISPR vectors and sgRNAs into cotyledonary explants [44].

Confirmation of transgene insertion and plant propagation should be conducted to confirm the editing events of T0 plants. To generate non-transgenic tomato lines, T-DNA must be segregated away by selfing T0 edited plants and cultivating the next generation (T1) plants. Further verification of the extent to which the genome of the edited plants suffers from off-target mutations due to the CRISPR/Cas9 system should be conducted. Among the tools that could be utilized to identify the possible off-target sites is the Cas-OFFinder online tool (http://www.rgenome.net/cas-offinder/, accessed 20 July 2024) [45].

Due to the features of simplicity in design and utilization, researchers have immediately adopted CRISPR/Cas as a versatile tool for genome editing [20,34,46]. Beyond Cas9, there are several other Cas proteins that are being explored for genome editing, Table 1. Each of these proteins offers unique characteristics that can be leveraged for different applications. Hillary and Ceasar reviewed in detail the different programmable sequence specificities of these proteins [47].

**Figure 1 plants-13-02269-f001:**
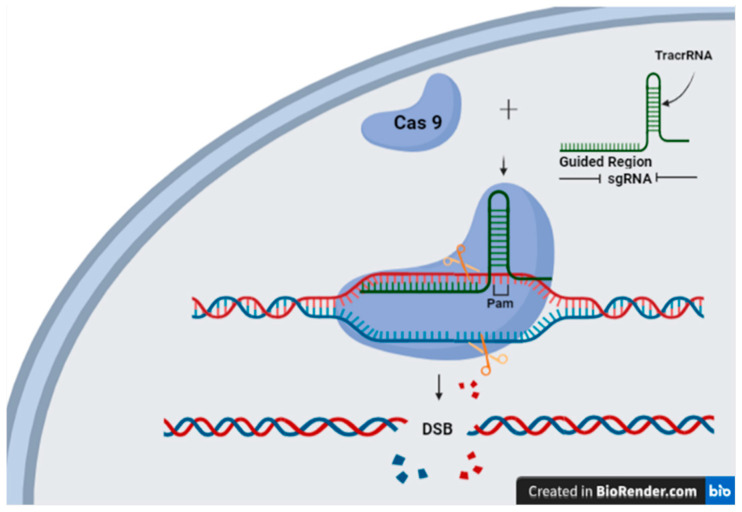
CRISPR-Cas9 mechanism is illustrated. The sgRNA consists of two parts: the guiding region and the scaffold region. The guiding region contains the complementary sequence of the targeted DNA (20–30 nucleotides), while the scaffold region includes the transactivating RNA that complexes with the sgRNA through hairpin loop structure formation. The steps that are involved in gene editing start with binding the Cas9 to the scaffold region of the sgRNA forming the CRISPR-Cas9 complex. The complex separates the double-stranded DNA near the PAM site (NGG, where N can be A, T, G, or C) after binding to it. Next, the sgRNA binds to its complementary target DNA sequence, and Cas9 induces a double-stranded break three base pairs upstream of the PAM site. The complex is released, and the break is repaired by cellular machinery such as non-homologous end joining (NHEJ) or homology-directed repair (HDR). Adapted from [48].

**Table 1 plants-13-02269-t001:** Features of the different types of Cas protein effectors that can be used with CRISPR technology for genome editing.

Type/Effector	Nuclease Domains	Target	Cut Structure	tracrRNA Requirement	PAM/PFS	References
II/Cas9	RuvC, HNH	dsDNA	blunt	Yes	3′ GC-rich PAM	[49]
V/CasX	RuvC	dsDNA	Staggered, 5′-overhangs	Yes	5′TTCN	[50,51]
V/Cas12	RuvC, NUC	dsDNA, ssDNA	Staggered, 5′-overhangs	No	5′-T-rich	[47,51]
V-A/Cas12a	RuvC, NUC	dsDNA	Staggered, 5′-overhangs	No	5′ AT-rich PAM	[47,49]
VI-A/Cas13a(C2c2)	2x HEPN	ssRNA	Guide-dependent RNA cuts + collateral RNA cleavage	No	3′ PFS: non-G	[49,52,53,54]
VI-B/Cas13b	2x HEPN	ssRNA	Guide-dependent RNA cuts + collateral RNA cleavage	No	5′ PFS: non-C; 30 PFS: NAN/NNA	
V/Cas14	RuvC	ssDNA	Collateral cleavage of ssDNA sequences	Yes	none	[47,55,56]

## 3. Applications of CRISPR/Cas9 in Editing Genes Related to Tomato Biotic Stress Resistance

Developing new genotypes resistant to biotic stresses is the best eco-friendly approach. However, classic breeding methods are time-consuming and labor-intensive. Genome editing via CRISPR/Cas9 has had a significant impact on breeding for biotic stresses in tomatoes. The ease of construction and its high efficiency have made CRISPR/Cas9 the most preferred tool for genome editing. Figure 2 provides an overview of the different biotic stresses and their corresponding tomato genes that have been manipulated using genome editing via CRISPR/Cas9.

### 3.1. CRIPSR/Cas9 Mediated Development and Understanding of Bacterial Resistance in Tomato

Several bacterial diseases of tomatoes are among the most destructive, affecting crops grown in both fields and greenhouses. The plant pathogen *Pseudomonas syringae* infects tomatoes and causes bacterial speck disease, which hinders their productivity, causing up to 52% loss of fruit weight [57]. *P. syringae* infection route involves entry through stomata, which are natural pores in plant leaves. Once *P. syringae* invades the plant, the initial immune response begins immediately by sealing the stomata to prevent bacterial entry. This process depends on the release of two hormones: salicylic acid (SA) and abscisic acid (ABA) [58,59]. However, several *P. syringae* strains, such as *Pto DC3000*, have developed a mechanism to alter this hormonal crosstalk by producing a mimic of the jasmonic acid (JA) hormone called coronatine (COR), which antagonizes the SA defense pathway [60]. When the JA pathway is activated, COR represses SA- dependent stomatal closure and facilitates bacterial invasion. This mechanism occurs when a functioning jasmonate Zim domain 2 (JAZ 2) co-receptor is present at the stomata. JAZ2 recognizes COR and prevents stomatal closure by interrupting the SA-dependent pathway, thus facilitating bacterial access to the plant and enabling its growth in the apoplast [60].

In order to manipulate the mode of action that COR-producing bacteria use to invade tomatoes, Ortigosa et al. [61] used CRISPR/Cas9 to edit the *SlJAZ2* gene, which is essential for regulating the dynamics of the stomata and produce *SlJAZ2Δjas* plants. Their results indicate that *Sljaz2Δjas* plants developed resistance against *P. syringae* by inhibiting the activity of phytotoxin COR at the stomata. This modification prevented stomatal re-opening by COR and the subsequent bacterial invasions through stomata. Mutant plants did not exhibit bacterial symptoms after infection, and this modification did not alter the stomatal aperture during transpiration. Furthermore, the mutant plants did not exhibit any alteration in their resistance to the necrotrophic fungal pathogen *B. cinerea*, the infectious agent of the tomato gray mold.

Another major and widespread disease is bacterial wilt, caused by the soil-borne bacterial pathogen *Ralstonia solanacearum*, which infects the tomato plant’s roots [62]. This disease is the most devastating and can cause yield loss by 50–100% annually. It is a typical vascular disease that harms the roots, stems, and leaves. The pathogen enters through wounds or mechanical injuries. When a plant is infected, the bacterial cells enter the xylem of the host roots to obtain the necessary nutrients, multiply, and kill the plant. In response to this stress, the host plant roots produce reactive oxygen species (ROS). Several genes enhance pathogen susceptibility by suppressing the expression of defense-related genes, known as negative regulatory genes [63]. *Hybrid Proline-rich Protein* (*HyPRP1*) and *differentially expressed in response to arachidonic acid 1* (*DEA1*) were identified as negative regulatory genes for several stresses [63,64]. To develop a bacterial wilt-resistant tomato genotype, CRISPR/Cas9 was used to knock out these negative regulatory genes, *HyPRP1* and *DEA1*. The edited plants exhibited resistance to the pathogen *R. solanacearum* [64]. ROS assays revealed a higher accumulation of ROS in wild-type (wt) plants compared to the mutant plants. In the same context, the group found a similar resistance response when the edited plants were infected with *Xanthomonas campestris* bacteria, the causal of bacterial spot disease.

Another approach to generate disease-resistant genotypes is to knock out the plant susceptibility genes, also known as S-genes, which increase the compatibility of plants with various pathogens [65]. Disabling S-genes generates broader- spectrum resistance compared to single resistance R-gene, thus offering great potential for plant breeding [66]. In *Arabidopsis*, the S gene *AtDMR6* encodes an enzyme that functions as a susceptibility factor to bacterial and oomycete pathogens [67]. Thomazella et al. characterized two orthologs of the *Arabidopsis* S gene *DMR6* in tomato plants, *SIDMR6-1* and *SIDMR6-2* [68]. However, only *SIDMR6-1* is up-regulated in response to pathogen infection and, therefore, plays a critical role in host plant immunity. The researchers further used CRISPR/Cas9 to induce mutations in the *SIDMR6-1* gene to increase tomato plant resistance. Two sgRNAs were designed to target specific exons of the *SIDMR6-1* gene, Causing a frameshift deletion and disrupting the SIDMR6 active site. The resulting mutants were *SIdmr6-1.1* and *SIdmr6-1.2*. Disease resistance assays were conducted with various bacterial pathogens. For instance, assays with *Pseudomonas syringae* demonstrated that the *SIdmr6-1.1* mutant line impaired pathogen growth and exhibited reduced disease severity compared to wt plants without any growth penalty. The enhanced pathogen resistance also led to increased levels of salicylic acid (SA), suggesting that pathogen resistance is associated with SA-mediated immune responses [68].

### 3.2. CRIPSR/Cas9 Mediated Development of Fungal Resistant Tomato

Fungal infections pose significant threats to tomato crops, affecting both yield and quality. CRISPR/Cas9 technology has emerged as a powerful tool in developing fungal-resistant tomato varieties by targeting and modifying specific genes involved in susceptibility and defense responses. This section delves into various strategies employed to enhance resistance against different fungal pathogens, highlighting the critical role of CRISPR/Cas9 in sustainable tomato cultivation.

#### 3.2.1. CRIPSR/Cas9 Mediated Development of Powdery Mildew Resistant Tomato

*Oidium neolycopersici* L. Kiss, a type of fungus that causes tomato powdery mildew disease, is an obligate biotroph that directly penetrates the tomato’s epidermal cells and forms a functional haustorium structure, which is crucial for successful fungal colonization on the leaf surface. Moreover, once the haustorium has matured, it starts drawing nutrients and water from the tomato plants. As a result, fungal colonies expand on the leaf surface due to new epiphytic hyphal growth. On the other hand, the invading pathogens secrete proteins from the haustoria to facilitate the delivery of fungal effector molecules into host cells, affecting the plant’s defenses and physiology [69]. 

The CRISPR/Cas9 system has been effectively utilized to increase the host plant’s resistance against tomato powdery mildew infection by eliminating the regulators that suppress the defense response of the plants and susceptibility genes. The mildew resistance locus O (MLO) susceptibility gene is the most commonly manipulated gene to acquire resistance against fungal diseases in plants [70]. According to Nekrasov et al. [71], tomato plants have 16 MLO genes, including *SlMlo1*, which serves as a key susceptibility gene to powdery mildew disease. They further targeted this gene to develop a tomato variety, Tomelo, resistant to the powdery mildew fungal pathogen using the CRISPR/Cas9 technology. Whole-genome sequencing proved that Tomelo is free of foreign DNA and only carries a deletion that is indistinguishable from naturally occurring mutations. Another group used the same technology to disrupt the *SlMlo1* gene, producing knockout mutants that are resistant to the powdery mildew-causing fungus *Oidium neolycopersici* and have no undesirable phenotypic effects [11].

Another investigation generated powdery mildew-resistant mutants using CRISPR/Cas9 by knocking-out one of the disease S-genes. *The powdery mildew-resistant* gene (*PMR4*) was found to be an *S*-gene in *Arabidopsis* as mutants that exhibit resistance to the disease infection [72]. Hubiers et al. confirmed that the tomato *gene Solyc07g053980* (*SlPMR4_h1*) has the highest level of homology with At*PMR4* and is most likely to have a similar function [73]. Martínez et al. [74] knocked out *SlPMR4*, and all the mutants that were inoculated with *O. neolycopersici* showed diminished susceptibility compared to the controls, as indicated by a lower disease index and significantly lower fungal biomass. Additionally, the primary haustoria percentage and the number of hyphae per infection unit were reduced in the mutants. These results support the finding that tomato *pmr4* mutants showed decreased susceptibility to *O. neolycopersici* but not complete resistance.

#### 3.2.2. Targeted CRISPR/Cas9 Editing of Susceptibility Genes to Enhance Resistance Late Blight Disease

The S-gene *PMR4* was further targeted for the resistance of tomato late blight (LB) disease [10]. LB disease, caused by the pathogen *Phytophthora infestans*, is a devastating disease and a serious concern for plant productivity [75]. Selected edited lines were inoculated with *P. infestans*, and four of them, fully knocked out at the *PMR4* locus, showed reduced disease symptoms (reduction in susceptibility from 55 to 80%) compared to control plants. 

Moreover, microRNAs (miRNAs), which are small noncoding RNAs, play a pivotal role in how plants respond to stress via regulatory genes [76]. The miRNAs are typically 20 to 24 nucleotides in length, and in tomatoes, mature *miR482b* and *miR482c* specifically have a length of 22 nucleotides. These miRNAs are crucial for the plant’s defense against pathogens. In a study conducted by Y. Hong et al. [77], the stem-loop structures of the precursor molecules *premiR482b* and *premiR482c*, which give rise to *miR482b* and *miR482c*, were disrupted using guide RNAs (gRNAs). This disruption led to the inhibition of *miR482b* and *miR482c* biogenesis and function, affecting the expression levels of genes within the same cluster. Consequently, the cultivars with CRISPR/Cas9-induced knockout of *miR482b* and *miR482c* exhibited enhanced resistance to *Phytophthora infestans* [77].

Interestingly, CRISPR/Cas9 was utilized to verify the role of different transcription factors that can be induced by *P. infestans* [78,79]. Liu et al. used CRISPR/Cas9 to confirm the role of the transcription factor *SlMYBS2*. Their results revealed that *SlMYBS2* acts as a positive regulator of the resistance to *P. infestans* infection by regulating the reactive oxygen species (ROS) level and the expression level of pathogenesis-related (*PR*) genes [78]. While Yang et al. [79] knocked out the transcription factor *SlbZIP68* via CRISPR/Cas9, the mutant plants exhibited a significant decrease in their resistant to the pathogen. They noticed reduced activity in defense enzymes and increased accumulation of ROS. Taken together, these findings demonstrate the role of transcription factors in enhancing resistance to pathogens.

#### 3.2.3. Targeting Genes for Fusarium Wilt Resistant

Fusarium wilt (FW) disease is ranked as the fifth most important fungal infection in tomatoes, resulting in a reduction in yield and fruit production of up to 80% in India. *Fusarium oxysporum* f. sp. *lycopersici* (Fol) causes FW infection through hyphae that invade and stele the apoplastic spaces and then block the xylem vessels, leading to lower development, leaf chlorosis, progressive wilting, and cell death [42,80].

During the FW infection in tomato plants, two regulatory genes are involved in suppressing the plant immune defense, including those encoding for Xylem sap protein 10 (XSP10) and Salicylic acid methyl transferase (SlSAMT). The *XSP10* gene encodes a protein acts as a compatibility factor for Fol, boosting Fol colonization in the tomato plant’s root. The other gene, *SlSAMT*, reduces the ability of the host plant to establish systemic resistance in response to pathogen infections by catalyzing the conversion of SA to methyl salicylate using *SAMT* enzymes [42]. Johni Debbarma et al. [42] utilized CRISPR/Cas9 technology to knock out these two genes individually and simultaneously. The results demonstrated that compared to single-gene editing, stable dual-gene CRISPR/Cas9 editing of *XSP10* and *SlSAMT* in disease-susceptible tomatoes showed significant FW resistance.

Another transcriptional factor mainly involved in regulating metabolic pathways, phytohormone signal transduction, response to biotic and abiotic stimuli, and other developmental processes, namely ethylene response element-binding protein (EREBP) [81,82]. A *Differential Display Tomato Fruit Ripening 10/A* (*DDTFR10/A*) gene, induced by ethylene and categorized in the EREBP family, its differential expression as a negative or positive regulator of transcriptional activity is still unknown [83]. Ijaz et al. [84] edited the *DDTFR10/A* gene, a susceptible tomato genotype using CRISPR/Cas9, and observed that FW tolerance in *DDTFR10/A* knocked out plants.

Proline-rich proteins (PRPs) are involved in cell-wall signaling, plant development, and stress responses [85]. *The hybrid proline-rich proteins 1* (*HyPRP1*) gene is a crucial gene of PRP and was shown to have diverse functions for stress responses of different plant species. *HyPRP1* positively regulates cell death and negatively regulates biotic stress defense in *Capsicum annuum* and *Nicotiana benthamiana* [63]. In tomatoes, *SlHyPRP1* was shown to be negatively involved in multi-stress responses (oxidative stress, dehydration, and salinity) [86]. Eliminating the *HyPRP1* protein’s active domains was reported to enhance resistance to salt stress in tomato [87].

Further investigation was conducted to check if a disabled *HyPRP1* gene will enhance resistance to biotic stress was conducted by [88]. Their results indicated that *SlHyPRP1*-edited genotypes increased susceptibility to *FOL* in two different tomato cultivars, and *SlHyPRP1* plays a positive role in regulating the plant defense against necrotrophic fungal pathogens. They further reported an opposite immune response against a hemibiotrophic bacterial pathogen when edited plants were infected with *P. syringae* (*Pto* DC3000).

In a study performed by Hanika et al. (2021) [89], the role of the *Walls Are Thin 1* (*WAT1*) gene in tomato plants was investigated. This gene facilitates the formation of secondary cell walls and the development of vascular tissues. Hanika et al. (2021) used CRISPR/Cas9 gene editing technology to create the *WAT1*-impaired plants. The *WAT1* knockout (*WAT1*-KO) tomato lines were created by designing and delivering CRISPR/Cas9 constructs that target the *WAT1* gene. The results of the study showed that the *WAT1*-KO plants exhibited growth defects and enhanced disease resistance, with the silencing of the *WAT1* gene causing severe growth abnormalities, including disrupted vascular development and dwarfism in tomato plants. However, the *WAT1*-silencing plants showed increased resistance to vascular wilt fungi, such as *Fusarium oxysporum* f. sp. lycopersici and *Verticillium dahliae*, despite the negative impacts on growth. An additional investigation demonstrated significant alteration in the cell wall composition of tomato plants following *WAT1* silencing, particularly an increase in lignin content and changes in gene expression related to cell wall production and modification. These factors are critical for plant defense against infections. Moreover, the findings suggest that the *WAT1*-silencing plants exhibited altered hormone signaling, including higher salicylic acid levels and higher expression of defense genes that are sensitive to salicylic acid. These changes likely contributed to the enhanced resistance observed in the silenced plants. Overall, the study’s results show that, despite the resulting growth abnormalities, modifying the *WAT1* gene can be a viable strategy for enhancing disease resistance in tomato plants [89].

#### 3.2.4. Enhancing Tomato Resistance to Grey Mold Disease

The increased susceptibility of tomato ripe fruit to fungal infection poses a devastating threat to crop production and marketability [90]. The gray mold plant disease, among the post-harvest pathogens in fruit, is caused by the common fungal phytopathogen *Botrytis cinerea* [91]. The ability of *B. cinerea* to produce conidia and show low host specificity leads to significant financial losses for more than 200 plant species, including tomato crops [92]. Brito et al. [91] reported that when ripe tomato fruit is affected by *B. cinerea*, significant post-harvest losses occur. During ripening, genes for cell wall degrading enzymes such as *Pectate lyases (PL)* and *polygalacturonase 2a (PG2a)* facilitate fruit softening. Mutants or silencing of these genes demonstrated the integrity of the cell wall in defense against *Botrytis cinerea* [93,94]. Silva et al. [95] targeted both genes via CRISPR/Cas9, the results showed that *PL* mutant fruit were nearly 30% firmer than fruit from the *PG2a* mutants and the wt lines. In conjunction with these firmness differences, RR fruit of the *PL* mutant lines, but not the *PG2a* mutants, exhibited reduced susceptibility to *gray mold* compared with the wt.

Furthermore, tomato *phospholipase Cs* (*SlPLCs*) gene activation is one of the earliest responses triggered by different pathogens regulating plant responses that, depending on the plant–pathogen interaction, result in plant resistance or susceptibility [96]. *SlPLC2* transcript levels increased upon xylanase infiltration (fungal elicitor), and *SlPLC2* participates in plant susceptibility to *B. cinerea* [97]. Perk et al. [98] obtained *SlPLC2* loss-of-function tomato lines more resistant to *B. cinerea* by means of CRISPR/Cas9 genome editing technology.

### 3.3. CRIPSR/Cas9 Mediated Development of Viral Resistant Tomato

There are 181 viral species infecting tomato crops. However, the major tomato viral pathogens spreading worldwide over the past 20 years include the following genera: *Begomovirus*, *Orthotospovirus*, *Tobamovirus*, *Potyvirus*, and *Crinivirus* [99,100,101]. Virus resistance breeding is the most promising method for controlling viral diseases [102]. Since plant viruses are highly dependent on the host to complete their replication cycle, the replication, assembly, and movement of viruses in plant cells require interaction with host–plant-specific factors, often referred to as susceptibility genes (*S* genes), for successful infection [103]. Editing these susceptible genes via CRISPR/Cas to break compatible interactions between the virus and host can help develop resistance in the host plants [104]. In this part, applications of CRISPR/Cas9 in tomato breeding for virus resistance are discussed.

#### 3.3.1. Targeting *TOM1* for Tomato Brown Rugose Fruit Virus Resistance

A new virus belonging to the genus *Tobamovirus* is called tomato brown rugose fruit virus (ToBRFV) [105]. ToBRFV overcomes the *tobamoviral* resistance gene *Tm-22* and is quickly spreading globally. The *TOBAMOVIRUS MULTIPLICATION1 (AtTOM1)* gene is necessary for the efficient multiplication of *tobamoviruses* in *Arabidopsis* [106]. Loss of function of the *TOM1* gene confers resistance to tomato against ToBRFV [106]. Moreover, the *TOM1* protein plays a role in the formation of active tobamoviral replication complexes and is a component of these complexes. CRISPR/Cas9-mediated multiplexed genome editing was utilized to knock out the four homologs of the *TOM1* gene simultaneously in tomatoes to confer ToBRFV resistance and also to generate plants with only three homologs knocked out for comparison [107]. In triple- or quadruple-mutant plants, the accumulation levels of *SlTOM1a*, *c*, or *d* mRNA from the mutant genes were lower than those from the non-mutated genes in triple mutants or from wt plants, probably due to nonsense-mediated mRNA decay. Furthermore, In *Sltom1* single or double mutants, ToBRFV coat protein (CP) accumulated to nearly wt levels. In some double mutants, including *Sltom1ac*, ToBRFV CP accumulation was slightly reduced compared with the wt plant. While in *Sltom1* triple mutants, ToBRFV CP accumulation was reduced compared with that in wt plants. The level of CP accumulation at 7 dpi was in the order wt > *Sltom1bcd* > *Sltom1abd* > *Sltom1abc* > *Sltom1acd*. Quadruple mutants showed resistance to the virus as no detectable symptoms were observed. These mutants also showed durable resistance to three other tobamoviral species. Furthermore, the knocked-out-mutant tomato plants did not experience any obvious growth defects under greenhouse conditions. Hence, *TOM1* homologs knockout is expected to have no apparent negative side effects, and to be useful to provide tobamoviral resistance to other plant species [107].

#### 3.3.2. Targeting *ELF4E* Gene for Potyvirus Resistance in Tomato

*Potyvirus* is a single-stranded RNA virus that affects tomato crops worldwide and is transmitted through aphids, contaminated seed, and plant materials mechanically and through wounds. Affected plants showed symptoms such as mosaic leaves and necrotic lesions. Upon infection, the virus will spread through the vascular system of the infected plant and interfere with physiological processes such as its ability to perform photosynthesis. Joung Yoon et al. [108] used CRISPR-Cas9 to induce allelic variation in the *EIF4E1* gene, a translation initiation factor to generate a Potyvirus-resistant cultivar. By designing a gRNA that targets mutations in the 5′ region or first exon of *EIF4E1* and causes non-functional protein formation. Progenies of the E0 transgenic plants have been shown to carry the *EIF4E1* mutation. The homozygous mutant lines have shown to be resistant to Pepper Mottle Virus (PepMoV). However, the mutants were susceptible to the Tobacco etch virus (TEV). That may be due to redundancy in *EIF4E1* and *EIF4E2*, allowing for viral infection. Knocking both genes out could potentially result in better TEV resistance [108].

#### 3.3.3. Targeting *SIPelo* Gene for Tomato Yellow Leaf Curl Virus Resistance

Tomato Yellow Leaf Curl Virus (TYLCV) is a circular single-stranded DNA virus of *Begomovirus*. TYLCV is one of the most detrimental viruses affecting tomato yield and causing abnormal leaf morphology, stunted growth, curled margins, and yellowing [11]. *SIPelo*, a susceptibility factor encoded by the host plant genome, synthesizes a messenger RNA surveillance factor known as Pelota (PELO).

Six TYLCV resistance quantitative loci (QTL) were identified recently, which include Ty-1, Ty-2, Ty-3, Ty-4, Ty-5, and Ty-6. The Ty-5 locus comprises the *SlPelo* gene which is further transcribed into mRNA known as the Pelota (PELO) protein that plays a crucial role in ribosome recycling during protein synthesis. Hence, knocking out the *SlPelo* gene restricts the proliferation of the TYLCV virus, causing viral resistance [11,108,109]. Pelo protein contains 387 amino acids and 3 eukaryotic translation termination factor 1 (ERF1), which are eRF1_1, eRF1_2, and eRF1_3. They revealed a role in protein synthesis and ribosome recycling. Thus, the CRISPR/Cas9 system was used to disturb the ERF1_1 domain in the PELO protein by inducing indels in the eRF1_1 domain coding region in the *SIPelo* locus that would eventually abolish the function of the PELO protein. The regenerated knockout SlPelo plants showed resistance to TYLCV infection. The loss-of-function of SlPelo restricted the TYLCV growth and did not produce any TYLCV symptoms in tested *SlPelo* mutant lines [11].

#### 3.3.4. Studying the Importance of Dicer-like Protein (*DCL*) Genes in Resistance to Mosaic Viruses Using CRISPR/Cas9 System

The RNA silencing system acts as an antiviral defense mechanism following its induction with virus-derived double-stranded RNAs. RNA silencing components are Dicer-like (DCL) nucleases, Argonaute (AGO) proteins, and RNA-dependent RNA polymerases (RDR) [110]. Dicer-like protein (DCL2b), a gene found in tomato crops, plays an important role in the plant’s defense system against Tomato mosaic virus (ToMV). DCL2b is involved in the biosynthesis of 22-nt small RNAs and the regulation of various plant genes involved in plant hormone signaling and mitochondrial metabolism. ToMV is a widespread and economically devastating viral disease that affects tomato crops. It is a ssRNA genome and belongs to the Tobamovirus genus. Symptoms of ToMV include browning of fruits, necrosis, and twisted and deformed leaves, which leads to decreased total fruit yield and quality. Furthermore, it can be transmitted directly through sap inoculation, contaminated tools, and indirectly through contaminated seeds [111].

Tomato is known to be targeted by two RNA viruses that may drastically reduce yields: tobacco mosaic virus (TMV) and potato virus X (PVX). In another investigation, Wang et al. [112] examined tomato dicer-like2 (*SlDCL2*) as it is also an essential component of resistance pathways against TMV and PVX [112]. They looked at the function that *DCL2* plays in this vulnerability. The scientists were able to investigate the effects of disrupting *DCL2* activity via CRISPR/Cas9. Interestingly, 22-nucleotide endogenous short RNAs, including a novel 22-nt miRNA termed miR6026, were found in the *dcl2a* and *dcl2b* mutant plants,. Subsequent investigation showed that miR6026 targets the *DCL2a*, *DCL2b*, and *DCL2d* mRNAs, resulting in a feedback loop that produces secondary small RNAs. The researchers were able to boost *DCL2* expression and improve resistance to TMV and PVX infections by expressing a miR6026 target mimic RNA to break this feedback loop. Upon infection with TMV or PVX, the *dcl2ab* mutant plants exhibited more severe viral symptoms in comparison to wt plants. Taken together, these results emphasize the critical function of DCL2 in antiviral defensive mechanisms. The work revealed a new miRNA-mediated regulatory pathway that involves DCL2 and emphasized the significance of this system for tomato defense against RNA viruses [112]. Furthermore, these studies detailed the importance of the DCL2b gene in the plant’s defense system.

### 3.4. CRIPSR/Cas9 Mediated Development of Weed Resistant Tomato

#### 3.4.1. Targeting CCD8 Gene in Tomato for *Phelipanche aegyptiaca* Weed Resistance

Among biotic stresses, weeds pose a significant economic threat to tomato crops worldwide, with all-season intervention capable of reducing marketable tomato yields by 36–92% [113]. *Phelipanche aegyptiaca* and *Orobanche spp.* are some of the most destructive challenges facing tomato production [114]. The pre-parasitic stage involves preparing the parasitic plant’s seeds for germination. The germination of these parasitic seeds depends on specific conditions, including a preconditioning process and exposure to strigolactones (SLs). SLs are chemical substances released by the roots of host plants. They are derived from carotenoids and involve three crucial genes: *More Axillary Growth 1* (*MAX1*), Carotenoid *Cleavage Dioxygenase 8 and 7* (*CCD8* and *CCD7*), as illustrated in (Figure 3). Unfortunately, effective methods to control parasitic weeds in most crops are limited. However, Bari et al. [115] stated that host resistance against parasitic weeds such as *P. aegyptiaca* can be achieved by using CRISPR/Cas9 to induce mutations in the *CCD8* gene. No apparent off-targets were found in a range of tomato lines with CCD8Cas9 mutations that varied in their *CCD8* insertions or deletions. The inserted *CCD8* mutations were inherited, according to genotype analysis of T1 plants. In comparison to the control tomato plants, the CCD8Cas9 mutant displayed morphological alterations such as dwarfing, excessive shoot branching, and adventitious root production. Furthermore, the SL-deficient CCD8Cas9 mutants showed a notable decrease in parasite infection as compared to non-mutant tomato plants, which were highly susceptible to parasite infestation [115,116,117].

#### 3.4.2. Targeting *MAX1* Gene in Tomato for *Phelipanche aegyptiaca* Weed Resistance

Another study conducted by Bari et al. investigated the possibility of using CRISPR/Cas9-mediated targeted editing of the tomato SL biosynthesis gene *MAX1* to confer resistance against the root parasitic weed *Phelipanche aegyptiaca.* The CRISPR/Cas9 technique was used to target the SL-biosynthesis gene *MAX1* (Solyc08g062950) in the tomato host plant. The Cas9/gRNA vector construct with an XmaI restriction site close to the protospacer adjacent motif (PAM) was designed to target the third exon of *MAX1* (positions 778–797 bp in the coding region). With no non-specific off-target impacts, the T0 plants were altered at the *MAX1* target site [115]. T0 line 1 was self-pollinated to produce T1 plants because T0 transgenic lines are often somatic. T1 plants demonstrated that the inserted mutations were persistently passed on to the following generations through a genotyping study. Notably, the morphological changes in *MAX1*-Cas9 heterozygous and homozygous T1 plants were identical and included increased axillary bud growth, decreased plant height, and adventitious root formation compared to the wt. The authors claimed that this performance was due to one normal copy of the gene being present and the other copy being mutated, which could affect the normal growth of plants since the protein may have a role in an important cellular function; however, these explanations need to be thoroughly investigated. Their findings demonstrate that the decreased SL (orobanchol) content in *MAX1*-Cas9 mutant lines results in resistance to the root parasitic weed *P. aegyptiaca*. In addition, compared to wt plants, there were alterations in the expression of the gene *PDS1* involved in the carotenoid biosynthesis pathway and the overall amount of carotenoid [115,116]. They concluded that genetic resistance to root parasitic weeds can be achieved utilizing CRISPR/Cas9 mediated targeted mutagenesis of the *MAX1*, an SL biosynthetic gene in tomatoes. An identical strategy could be effectually used against other parasitic weeds to establish host resistance [115].

### 3.5. Enhancing Herbicide Resistance in Tomato via CRISPR/Cas9

Weeds rapidly increase their resistance to herbicides, creating a global challenge to food security. Weeds competing with tomatoes can cause yield reduction of up to 92% and reduce fruit quality, which decreases their market value [113,118,119]. For modern large-scale crop production, using herbicides is the main approach used for weed control, especially with the modern systems of large-scale crop production.

The development of herbicide-resistant *S. lycopersicum* is of economic importance, reducing labor and costs required for large-scale production. With this aim, Yang et al. [120] tested different gRNAs to edit herbicide-binding proteins encoding genes, *phytoene desaturase (pds)*, acetolactate synthase (*ALS*), and 5-enolpyruvylshikimate-3-phosphate synthase *(EPSPS)*, in tomato (*S. lycopersicum* cv. Micro-Tom) using CRISPR/Cas9. Successfully, they were able to generate ALS-mutant plants. The *ALS* gene encodes the enzyme that catalyzes the first step of the biosynthetic pathway for branched-chain amino acids (BCAAs), specifically for the production of valine, leucine, and isoleucine. Several investigations have revealed that point mutations in this gene can provide dominant resistance to ALS inhibitors. Targeting the *ALS* gene constitutes a tool of choice for the selection of edited plants [121,122]. Proline-197 residue (amino acid number standardized to the *Arabidopsis* sequence, corresponding to Proline-186 in tomato and potato *ALS1*) mutation is one of the most commonly reported mutations providing chlorsulfuron resistance. Viellete et al. [123] used cytidine base editors (CBEs) for precise plant gene editing. They designed one sgRNA to target the *SlALS1* gene, considering that Pro186 (CCA) is located in the editing window of Target-AID (target-activation-induced cytidine deaminase). After kanamycin selection pressure during the Agrobacterium transient expression stage, the plant tissues were transferred to a selective medium that contained 40 ng mL^−1^ chlorsulfuron. The selective medium allowed only the *SlALS1* locus mutated cells to grow and regenerate plantlets. Out of these plants, 71.4% were edited at the Pro186 (CCA) codon, and 98.7% of them were modified at the C-14 position. Of these C-14 positions, 80% were C to T changes, with 16.7% being homozygous mutants [123].

## 4. De Novo Domestication and Multiplex Genome Editing in Tomato for Biotic Stress Resistance

Innovations in CRISPR/Cas9 provide precise editing of multiple targeted genes simultaneously through multiplexing to accelerate breeding at reduced costs [20,124]. De novo domestication involves a suite of loci that are involved in the reshaping of the morphological and agronomical characters of the wild species to create a novel crop. Wild species harbor beneficial traits such as disease resistance and stress tolerance. Zsögön et al. and Rodriguez-Leal are among the very early multiplexing studies conducted with the aim of de novo domestication of tomatoes. These studies provide an example for a specific class of De Novo Domesticated genome-edited tomato lines targeting improved plant architecture flower and fruit traits, which are inherited after editing the parental wild plant species [21,124].

Multiplexing is conducted by a toolbox designed to insert multiple sgRNA expression cassettes into a single binary vector. This approach is based on the principle that several sequential rounds of standard cloning steps can be utilized to introduce different expression cassettes containing sgRNAs targeting different genes into a single construct. However, this classic basic method is very laborious, and there is a limited number of assembled sgRNAs [125,126,127,128]. A more advanced technique, called Golden Gate cloning, depends on the use of type IIS restriction enzymes in the DNA assembly. This technique facilitates the cloning of different constructs into binary vectors based on the different toolkits available for CRISPR/Cas9-mediated genome editing. These toolkits are either based on assembling multiple sgRNA expression cassettes, each transcribed from RNA polymerase III promoters such as U6 or U3 [129,130,131], or through a single transcript of polycistronic mRNAs that are cleaved into individual gRNAs post-transcriptionally by the naturally present in the host cells [132], CRISPR-associated RNA endoribonuclease Csy4 from *Pseudomonas aeruginosa* [133]; the tRNA processing enzymes; or self-processing ribozymes [134,135].

Debbarma et al. [42] provided a comprehensive downstream analysis of the two *S* genes *Xylem sap protein 10 (XSP10) and salicylic acid methyle transferase (SlSAMT)* via CRISPR/Cas9-mediated editing of single (*XSP10* and *SlSAMT* individually) and dual-gene (*XSP10* and *SlSAMT* simultaneously). The dual-gene CRISPR-edited lines (CRELs) of *XSP10* and *SlSAMT* at GE_1_ generation conferred a strong phenotypic tolerance to FW disease compared to single-gene-edited lines [42]. While Pramanik et al. [11] conducted multiplex editing for *Pelo* and *Mlo1* genes, they did not provide the response of the dual mutants. In another investigation, multiplex editing of *SlHyPRP1* and *SlDEA1* genes in plants might have imparted an enhanced immunity of the plants, which, in turn, caused less PAMP-triggered immunity (PTI), whereas programmed cell death is an effector-triggered immunity (ETI) production [64]. This enhanced immunity against bacterial leaf spots caused by *Xanthomonas campestris*, thereby generating less damage to the edited plant leaves as compared to wt leaves. Similarly, the edited plants exposed to *R. solanacearum* pathogen showed a clear difference in disease progression as compared with wt-treated plants. Moreover, wt-treated roots showed significantly higher accumulation of ROS than the edited plant roots.

## 5. Conclusions and Future Perspective

The use of genome editing techniques holds great promise for the breeding of tomato plants in the future and for their quick adaptation to biotic stressors. The manipulation of specific genes that render tomato plants susceptible to pests and pathogens has demonstrated very promising results (Table 2). Furthermore, the development of these tools has revolutionized genome editing and tomato breeding. These tools allowed researchers to make highly effective modifications that can enhance the tomato’s disease resistance and innate defense response mechanisms that can be inherited by newer generations. These modifications allowed researchers to develop measures to treat a wide range of biotic stresses, including fungal, viral, and weed, that pose a threat to tomato plants. These tools modify the target plant’s genome without introducing foreign DNA, minimizing various regulatory concerns and facilitating the approval of this method to address the urgent need for resistant tomato crops. It has allowed researchers to make highly effective modifications that enhance the tomato’s resistance and innate defense response mechanisms that can be conferred to newer generations. Although genome editing is very promising, there are various limitations that are yet to be met, including off-target effects, cost, complexity, and potential future consequences [136,137]. However, several online bioinformatics tools are being developed to be utilized to select the most optimum sequence and minimize off-target events. Continued research and development of these tools will provide a promising solution to reduce yield losses, maintain healthy crops, and ensure food security. Innovations in Cas9 offer precise editing of targeted genes and even simultaneous editing of multiple genes by multiplexing to accelerate breeding at reduced costs. Multiplex genome editing could facilitate the improvement of multiple traits simultaneously, as noted by Abdelrahman et al. [20]. Recently, Zsögön et al. [124] succeeded in de-novo domesticating wild tomatoes using CRISPR/Cas9 mediated multiplex editing for six loci that are important for yield and productivity. This technique is considered a breakthrough by plant breeders, as it makes use of the wild beneficial traits such as biotic and abiotic stress resistance.

Further research will undoubtedly refine and optimize genome editing tools such as prime, twin prime, and PASTE approaches, which are rarely mentioned in recent studies on tomatoes and the approaches of using these tools to modify tomatoes. More research needs to be conducted in order to address the safety concerns of genome editing, such as minimizing off/on-target effects, diminishing integration of CRISPR/Cas components or plasmid DNA, reducing costs, and improving efficiency [136,137]. Moreover, the refinement of these tools will allow for more reliable and precise genome modification to combat biotic stresses. Gaining the trust of authorities, farmers, and the public is considered essential by researchers and stakeholders. As public awareness is also necessary for the acceptability of genome-edited genotypes, which no longer maintain any foreign genes in their genetic background, governmental and societal concerns must continue to be addressed. Furthermore, ethical considerations and responsible practices must still be maintained. Even though genome editing methods have been acknowledged and accepted in more than several countries, including the US, Australia, and Canada, other countries, including the EU, New Zealand, Japan, Norway, and Switzerland, have started revising their policies. Furthermore, Japan has recently introduced specific legislation to address genome-edited organisms [138,139]. In 2021, Japan approved two genome-edited fishes were approved for commercial sale and a genome-edited tomato variety with an increased content of gamma-aminobutyric acid (GABA tomato) [138,139]. The advances in CRISPR/Cas9 will contribute to the advancement of agriculture production that will contribute to food security.

**Table 2 plants-13-02269-t002:** A list of genes targeted with genome editing to enhance different biotic resistance in tomatoes.

Infection	Disease	Editing Tool	Target	Editing Method	Effect	References
Bacterial	Bacterial speck	CRISPR/Cas9	*SlJAZ2*	Knockout	Bacterial speck resistance in tomato	[61]
	Bacterial wilt disease	RNA- interference (RNAi)	*DPS*	-----------	Soil-borne pathogen resistance in root	[28]
Herbicidal	Chlorsulfuron	Base editing (CBE)	*SlALS1*	-----------	Chlorsulfuron resistant tomato	[123]
Fungal	Powdery mildew	CRISPR/Cas9	*SlMlo1*	Knockout	powdery mildew-resistant tomato	[70,71]
			*Pmr4*			[140]
	Fusarium wilt		*CYCLOPS/IPD3*		Fusarium wilt-resistant tomato	[141]
			*XSP10 and SAMT*		Multiplexing both genes showed significant Fusarium wilt resistance	[42]
	Late blight		*miR48b*		Late blight-resistant tomato	[77]
			*miR482c*			
	Gray mold		*S1PL*		Reduced susceptibility of tomato by >50%.	[95]
Viral	Tomato brown rugose fruit virus resistance (ToBRFV)	CRISPR/Cas9	*TOM1*	Knockout	ToBRFV resistant tomato	[107]
	Tomato Yellow Leaf Curl Virus (TYLCV)		*SlPelo*		Decreases accumulation of TYLCV	[11]
	Potyvirus		*ELF4E*		Conferred resistance to one type of potyvirus, Pepper Mottle Virus (PepMoV), but was susceptible to Tobacco etch virus (TEV)	[108]
	Tomato mosaic virus	-----------	*SlDCL2b*	------------	--------------	[111]
	Potato virus X	-----------	*SlDCL2a*			[112]
			*SlDCL2b*			
Weed	Broomrapes	CRISPR/Cas9	*CCD8*	Knockout	Remarkable reduction in parasite infection	[117]
	Root parasitic weed		*MAX1*		Root parasitic weed-resistant tomato	[115]

## Figures and Tables

**Figure 2 plants-13-02269-f002:**
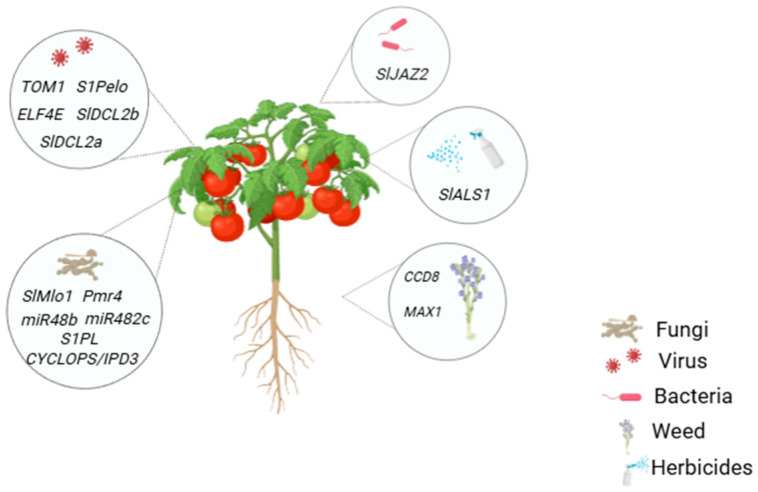
Demonstration of several types of possible biotic stresses that might infect the tomato plant and the genes that were targeted in the tomato in order to induce resistance against those biotic stresses.

**Figure 3 plants-13-02269-f003:**
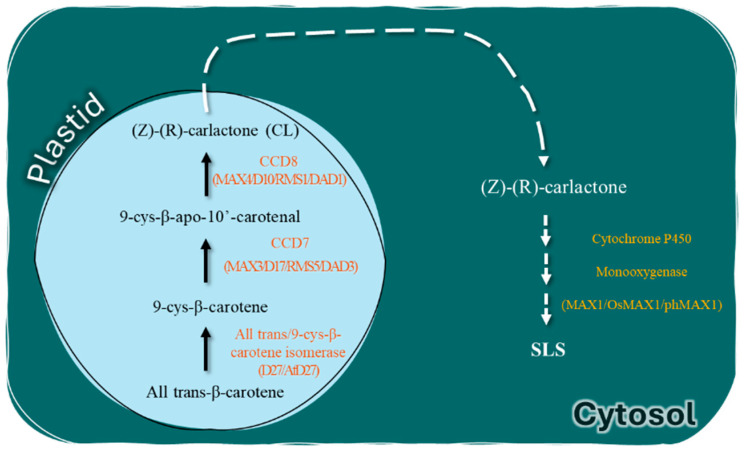
The figure depicts a pathway diagram illustrating the biosynthesis of SLs in plants. The pathway begins with the conversion of all-trans-β-carotene to carlactone (CL) by the enzyme DWARF 27 (D27). Two additional enzymes, *CCDs* 7 and 8, act sequentially on CL. *MAX1* then oxidizes CL to produce a variety of SLs. Figure was adapted from [116].

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
