# Peer review of "Revolutionizing Tomato Cultivation: CRISPR/Cas9 Mediated Biotic Stress Resistance"

_plants, 2024, doi:10.3390/plants13162269_

Round 1
Reviewer 1 Report
Comments and Suggestions for Authors
Dear authors
The manuscript provides an overview on the development of genome edited tomato plants with increased resistance to a variety of biotic stressors, including bacterial, fungal and viral plant pathogens as well as the generation of herbicide resistant tomato varieties for an improved management of weeds in tomato crops with herbicides.
The manuscript would benefit from a comprehensive revision to improve some elements of the structure of the discussion and in particular a thorough editorial revision to enhance the intellegibility of the language.
I will provide some suggestions for revision in the detailed comments below, but also want to indicate a number of general points for revision:
1) In the last para of the introduction the manuscript claims that this is the first attempt to review the advances in tomato resistance to biotic stresses using genome editing tools. This does not seem to be entirely correct. There are a number of recent papers, which address this topic; some, however, in broader contexts of e.g. advances in tomato development with genome editing regarding several or multiple breeding objectives (including biotic stress) or advances in genome editing in crop plants including tomato and biotic stress resistance as subtopics. Two of these papers (Salava et al. 2021; Amoroso et al. 2022) are actually referenced in the manuscript, however in different contexts; others are not (Chandrasekaran et al. 2021; Kumar et al. 2022; Li et al. 2022). It may be good to revise the above statement and include some other relevant references (maybe pointing out the specific focus of the manuscript at hand).
Furthermore, the introduction refers to “genomic tools” with no definition provided. In the manuscript at hand these don´t seem to include genome editing approaches, which would be confusing considering the common use of this term.
2) The treatment of the CRISPR/Cas technology is short and concise, however, it may be good to include some references to papers which provide a more detailed discussion of applications of this technology in plant biotech.
Also pls mind that you use both gRNA (guideRNA) and sgRNA (single guide RNA) interchangably in the text and the figures (such as Fig1) – pls. use terms consistently throughout.
3) The headline of Chapter 3.3 is missing – pls. revise and include a heading!
4) Chapter 4 is discussing “weed resistant tomato”, however, the chapter actually addresses two different types of applications: (i) genome-edited tomato lines with an increased resistance against colonialization by parasitic plants (or parasitic weeds), which should not be confused with “agricultural weeds” in a common sense and (ii) herbicide-resistant genome-edited tomato lines which increase the range of chemical herbicides which may be used for the agricultural management of tomato cultures (these tomato lines are not weed-resistant in a strict sense). I suggest to discuss these applications with regard to their respective characeristics in different sections of the manuscript and not under the common heading “weed resistant tomato”.
5) The paper by (Zsögön et al. 2018) is refered to only as an example for multiplex editing without reference that this is actually an example for a specific class of De Novo Domesticated genome edited tomato lines with an increased resistance to biotic stress (plant diseases) which is inherited from the modified wild plant species. I suggest to discuss this type of application as an approach that is different to the other described applications in a separate section in chapter 3! In such approaches the genome-edited traits are not directly responsible for the higher resistance towards biotic stressors, however the edited lines display an increased capacity to cope with biotic und abiotic stress due to their genetic background that is derived from a wildform rather than from a highly domesticated, and more suszeptible tomato line. Such approaches in tomato are described in various papers (Zsögön et al. 2018; Li et al. 2018), and have relevant consequences regarding the biosafety considerations which are required for such types of genome-edited plants (Eckerstorfer et al. 2023).
6) Safety considerations are mentioned in the chapter 4, however without any references. Such aspects are discussed in general by (Agapito-Tenfen et al. 2018; Duensing et al. 2018; Eckerstorfer et al. 2023; Kawall et al. 2020), off-target effects are discussed e.g. by (Chu und Agapito-Tenfen 2022).
7) As regards editorial revisions species names should be typeset consistently in Italics; pls. amend throughout including at: L131, L149, L211, L244, L255, L258, L267, L282, L295, L399, L402, L435, L438, L463, L467.
8) Some sentences need to be thoroughly revised with regard to formulations and grammar to improve intelligability and readability, pls. amend among others: L53-55, L68-71, L82-84, L86-88, L135-138, L146-148, L174, L209-218,L233-237, L258-262, L284-290, L303-306, L326-329, L335-337, L349-350, L382-392, L400-402, L406-412, L443-446, L4475-478, L492-496, L513-518.
Detailed comments
Abstract
L17: Consider deleting “However,” in the sentence
L19: Pls. revise wording; I suggest to change to
..but this approach requires significant time and efforts.
L22: Revise to: … precise, minute adjustments in the modified genes without …
L26: Revise expression “biotic resistant tomato lines” – this expression is used throughout the manuscript, I suggest changing to: tomato lines with resistance to biotic stress
Introduction
L30: Pls. change to:
Tomato (Solanum lycopersicon L.) is considered as one of the most important vegetable crops with an annual production of 182.3 million tons.
L32: Second highest of what? Pls. include explanation
L35: Revise sentence, I suggest:
Unfortunately, this increase in demand must be met with the current limited land and water resources.
L41: Revise to: “also in changes of plant physiology,..”
L47: Pls. revise (“adaptable pathogens”?)
L60: Check reference 14 – this is a publication dealing with genome editing in tomato not with genetic engineering based on classical transformation techniques. If genome editing is referred to this would be result in contradictions with the next sentences!
2. CRISPR/Cas9 technology
L83: Explain (What is this?) or revise: “detailed achievements-based CRISPR/Cas9”
L97: Change “correct” to: "a selected" editing site
L98: Revise sentence (I suggest: Subsequently, the double-strand breaks in the target DNA are repaired via one of two pathways: …)
L104: If the figure is taken from literature pls provide the reference!
Ch. 3
L122: Change “toll” to “tool” and revise the rest of the sentence!
L131: Revises “relies on..”
L136: Delete “As”
L143: Revise to: … and and prevents closing of the stomata by ..
L160: Suggest “produce” instead of “emit”
L179: delete “it”
L193. Suggest to use “disrupt” or “disable” instead of “to cut out”
L200: Revise to: “which resulted in an inceased…”
L221: Consider “important” instead of “catastrophic”
L222: Revise to: “ … in the yield and fruit production of up to 80% in India.”
L231: Delete “acquired”
L244: Delete “that
L247: Change “which” to “and”
L252-253: Delete ” It is also worth mentioning that” and explain which genes are affected
L262: Consider “identify” instead of “comprehend”
L270: Change “over” to “more than”
L273: Consider the following revision:
In contrast to predictions, this proved that mature fruit does not have an insufficient immune response that …
makes it incapable of preventing the growth of fungi [51
L276: Delete “must”
L297: Delete “unequivocally”
L310: Revise: “outcompetes” (overcomes?)
L311: Change “gaining ground” for “ spreading”
L317: Are there any relevant results for the comparator lines which should be mentioned?
L318: Change “Quadrable” to “Quadruple”
L329: Pls. revise to:
… the virus will spread through the vascular system of the infected plants …
L341: Revise expression: “…DNA virus of begomovirus.”
L342: Change to: “…viruses affecting tomato yield and causing abnormal leaf morphology, stunted growth, curled margins, and yellowing [9].”
L358: Reconsider heading: other headings indicate targeted virus
L372: Delete “specific”
L377: Revise the sentence
L378: Delete “set out to” and change examine to “examined”
L409-418: This description seems to be too technical and detailed.
L443: Delete “very well” and “matured and”
L446-448: Explain why heterozygous and homozygous plants have similar phenotypes.
L460-462: Delete “subsequently” and change to “ for modern large-scale crop production.”
L382. Pls. explain or revise:“commercially significant RNA viruses”?
L484-486: Change to:
Out of these plants 71.4% were edited at the Pro186 (CCA) codon, 98.7 % of them were modified at the C-14 position. Of these C-14 positions, 80% were C to T changes, 16.7% being homozygous mutants.
Conclusions
L490: Change to: “specific genes that render tomato plants susceptible to pests and pathogens..”
L505: revise to “in de-novo domestication” – Consider comments (5) above for revision!
L509: Revise “toomato”
Literaturverzeichnis
Agapito-Tenfen, Sarah Z.; Okoli, Arinze S.; Bernstein, Michael J.; Wikmark, Odd-Gunnar; Myhr, Anne I. (2018): Revisiting Risk Governance of GM Plants: The Need to Consider New and Emerging Gene-Editing Techniques. In: Frontiers in plant science 9, S. 1874. DOI: 10.3389/fpls.2018.01874.
Amoroso, Ciro Gianmaria; Panthee, Dilip R.; Andolfo, Giuseppe; Ramìrez, Felipe Palau; Ercolano, Maria Raffaella (2022): Genomic Tools for Improving Tomato to Biotic Stress Resistance. In: Chittaranjan Kole (Hg.): Genomic Designing for Biotic Stress Resistant Vegetable Crops. 1st ed. 2022. Cham: Springer International Publishing; Springer, S. 1–35. Online verfügbar unter https://link.springer.com/chapter/10.1007/978-3-030-97785-6_1.
Chandrasekaran, Murugesan; Boopathi, Thangavelu; Paramasivan, Manivannan (2021): A status-quo review on CRISPR-Cas9 gene editing applications in tomato. In: International Journal of Biological Macromolecules 190, S. 120–129. DOI: 10.1016/j.ijbiomac.2021.08.169.
Chu, Philomena; Agapito-Tenfen, Sarah Zanon (2022): Unintended Genomic Outcomes in Current and Next Generation GM Techniques: A Systematic Review. In: Plants (Basel, Switzerland) 11 (21), S. 2997. DOI: 10.3390/plants11212997.
Duensing, Nina; Sprink, Thorben; Parrott, Wayne A.; Fedorova, Maria; Lema, Martin A.; Wolt, Jeffrey D.; Bartsch, Detlef (2018): Novel Features and Considerations for ERA and Regulation of Crops Produced by Genome Editing. In: Frontiers in bioengineering and biotechnology 6, S. 79. DOI: 10.3389/fbioe.2018.00079.
Eckerstorfer, Michael F.; Dolezel, Marion; Engelhard, Margret; Giovannelli, Valeria; Grabowski, Marcin; Heissenberger, Andreas et al. (2023): Recommendations for the Assessment of Potential Environmental Effects of Genome-Editing Applications in Plants in the EU. In: Plants (Basel, Switzerland) 12 (9). DOI: 10.3390/plants12091764.
Kawall, Katharina; Cotter, Janet; Then, Christoph (2020): Broadening the GMO risk assessment in the EU for genome editing technologies in agriculture. In: Environ Sci Eur 32 (1). DOI: 10.1186/s12302-020-00361-2.
Kumar, S. Anil; Kottam, Suman Kumar; Narasu, M. Laxmi; Kumari, P. Hima (2022): Recent Trends in Targeting Genome Editing of Tomato for Abiotic and Biotic Stress Tolerance. In: S. H. Wani und G. Hensel (Hg.): Genome editing : current technology advances and applications for crop improvement: Springer, S. 273–285. Online verfügbar unter https://link.springer.com/chapter/10.1007/978-3-031-08072-2_15.
Li, Tingdong; Yang, Xinping; Yu, Yuan; Si, Xiaomin; Zhai, Xiawan; Zhang, Huawei et al. (2018): Domestication of wild tomato is accelerated by genome editing. In: Nat Biotechnol. DOI: 10.1038/nbt.4273.
Li, Yangyang; Wu, Xiuzhe; Zhang, Yan; Zhang, Qiang (2022): CRISPR/Cas genome editing improves abiotic and biotic stress tolerance of crops. In: Frontiers in genome editing 4, S. 987817. DOI: 10.3389/fgeed.2022.987817.
Salava, Hymavathi; Thula, Sravankumar; Mohan, Vijee; Kumar, Rahul; Maghuly, Fatemeh (2021): Application of Genome Editing in Tomato Breeding: Mechanisms, Advances, and Prospects. In: International Journal of Molecular Sciences 22 (2), S. 682. DOI: 10.3390/ijms22020682.
Zsögön, Agustin; ÄŒermák, Tomáš; Naves, Emmanuel Rezende; Notini, Marcela Morato; Edel, Kai H.; Weinl, Stefan et al. (2018): De novo domestication of wild tomato using genome editing. In: Nat Biotechnol. DOI: 10.1038/nbt.4272.
Comments on the Quality of English Language
see general and detailed comments above
Author Response
Dear authors
The manuscript provides an overview on the development of genome edited tomato plants with increased resistance to a variety of biotic stressors, including bacterial, fungal and viral plant pathogens as well as the generation of herbicide resistant tomato varieties for an improved management of weeds in tomato crops with herbicides.
The manuscript would benefit from a comprehensive revision to improve some elements of the structure of the discussion and in particular a thorough editorial revision to enhance the intelligibility of the language.
Respected Prof.
We would like to thank you for the valuable comment and suggestions which are highly appreciated. All the manuscript was revised based on the comments and suggestions. Also, whole the manuscript was revised and enhanced the intelligibility of the language. Please find below our responses.
I will provide some suggestions for revision in the detailed comments below, but also want to indicate a number of general points for revision:
1) In the last para of the introduction the manuscript claims that this is the first attempt to review the advances in tomato resistance to biotic stresses using genome editing tools. This does not seem to be entirely correct. There are a number of recent papers, which address this topic; some, however, in broader contexts of e.g. advances in tomato development with genome editing regarding several or multiple breeding objectives (including biotic stress) or advances in genome editing in crop plants including tomato and biotic stress resistance as subtopics. Two of these papers (Salava et al. 2021; Amoroso et al. 2022) are actually referenced in the manuscript, however in different contexts; others are not (Chandrasekaran et al. 2021; Kumar et al. 2022; Li et al. 2022).
Modified and the references were added.
It may be good to revise the above statement and include some other relevant references (maybe pointing out the specific focus of the manuscript at hand).
Modified and we pointed that this manuscript focuses specifically on biotic stress.
Furthermore, the introduction refers to “genomic tools” with no definition provided. In the manuscript at hand these don´t seem to include genome editing approaches, which would be confusing considering the common use of this term.
Modified to several approaches.
2) The treatment of the CRISPR/Cas technology is short and concise, however, it may be good to include some references to papers which provide a more detailed discussion of applications of this technology in plant biotech.
This part was optimized and references discussing the application in details were added.
Also pls mind that you use both gRNA (guideRNA) and sgRNA (single guide RNA) interchangably in the text and the figures (such as Fig1) – pls. use terms consistently throughout.
Done. We changed all to sgRNA
3) The headline of Chapter 3.3 is missing – pls. revise and include a heading!
Done.
4) Chapter 4 is discussing “weed resistant tomato”, however, the chapter actually addresses two different types of applications: (i) genome-edited tomato lines with an increased resistance against colonialization by parasitic plants (or parasitic weeds), which should not be confused with “agricultural weeds” in a common sense and (ii) herbicide-resistant genome-edited tomato lines which increase the range of chemical herbicides which may be used for the agricultural management of tomato cultures (these tomato lines are not weed-resistant in a strict sense). I suggest to discuss these applications with regard to their respective characeristics in different sections of the manuscript and not under the common heading “weed resistant tomato”.
Done. Sections were separated into: 3.4. weed resistant and 3.5. herbicide resistance
5) The paper by (Zsögön et al. 2018) is refered to only as an example for multiplex editing without reference that this is actually an example for a specific class of De Novo Domesticated genome edited tomato lines with an increased resistance to biotic stress (plant diseases) which is inherited from the modified wild plant species. I suggest to discuss this type of application as an approach that is different to the other described applications in a separate section in chapter 3! In such approaches the genome-edited traits are not directly responsible for the higher resistance towards biotic stressors, however the edited lines display an increased capacity to cope with biotic und abiotic stress due to their genetic background that is derived from a wildform rather than from a highly domesticated, and more suszeptible tomato line. Such approaches in tomato are described in various papers (Zsögön et al. 2018; Li et al. 2018), and have relevant consequences regarding the biosafety considerations which are required for such types of genome-edited plants (Eckerstorfer et al. 2023).
Done. We added a new part for multiplex genome editing for biotic stress resistance in tomato.
6) Safety considerations are mentioned in the chapter 4, however without any references. Such aspects are discussed in general by (Agapito-Tenfen et al. 2018; Duensing et al. 2018; Eckerstorfer et al. 2023; Kawall et al. 2020), off-target effects are discussed e.g. by (Chu und Agapito-Tenfen 2022).
The references were added.
7) As regards editorial revisions species names should be typeset consistently in Italics; pls. amend throughout including at: L131, L149, L211, L244, L255, L258, L267, L282, L295, L399, L402, L435, L438, L463, L467.
Done.
8) Some sentences need to be thoroughly revised with regard to formulations and grammar to improve intelligability and readability, pls. amend among others: L53-55, L68-71, L82-84, L86-88, L135-138, L146-148, L174, L209-218,L233-237, L258-262, L284-290, L303-306, L326-329, L335-337, L349-350, L382-392, L400-402, L406-412, L443-446, L4475-478, L492-496, L513-518.
Done
Detailed comments
Abstract
L17: Consider deleting “However,” in the sentence
Done.
L19: Pls. revise wording; I suggest to change to
..but this approach requires significant time and efforts.
Done.
L22: Revise to: … precise, minute adjustments in the modified genes without …
Done.
L26: Revise expression “biotic resistant tomato lines” – this expression is used throughout the manuscript, I suggest changing to: tomato lines with resistance to biotic stress
Done.
Introduction
L30: Pls. change to:
Tomato (Solanum lycopersicon L.) is considered as one of the most important vegetable crops with an annual production of 182.3 million tons.
Done.
L32: Second highest of what? Pls. include explanation
Done.
L35: Revise sentence, I suggest:
Unfortunately, this increase in demand must be met with the current limited land and water resources.
Done.
L41: Revise to: “also in changes of plant physiology,..”
Done.
L47: Pls. revise (“adaptable pathogens”?)
Done.
L60: Check reference 14 – this is a publication dealing with genome editing in tomato not with genetic engineering based on classical transformation techniques. If genome editing is referred to this would be result in contradictions with the next sentences!
Done.
- CRISPR/Cas9 technology
L83: Explain (What is this?) or revise: “detailed achievements-based CRISPR/Cas9”
Done.
L97: Change “correct” to: "a selected" editing site
Done.
L98: Revise sentence (I suggest: Subsequently, the double-strand breaks in the target DNA are repaired via one of two pathways: …)
Done.
L104: If the figure is taken from literature pls provide the reference!
Done.
Ch. 3
L122: Change “toll” to “tool” and revise the rest of the sentence!
Done.
L131: Revises “relies on..”
Done.
L136: Delete “As”
Done.
L143: Revise to: … and and prevents closing of the stomata by ..
Done.
L160: Suggest “produce” instead of “emit”
Done.
L179: delete “it”
Done.
L193. Suggest to use “disrupt” or “disable” instead of “to cut out”
Done.
L200: Revise to: “which resulted in an inceased…”
Done.
L221: Consider “important” instead of “catastrophic”
Done.
L222: Revise to: “ … in the yield and fruit production of up to 80% in India.”
Done.
L231: Delete “acquired”
Done.
L244: Delete “that
Done.
L247: Change “which” to “and”
Done.
L252-253: Delete ” It is also worth mentioning that” and explain which genes are affected
Done.
L262: Consider “identify” instead of “comprehend”
Done.
L270: Change “over” to “more than”
Done.
L273: Consider the following revision:
In contrast to predictions, this proved that mature fruit does not have an insufficient immune response that …
makes it incapable of preventing the growth of fungi [51
Done.
L276: Delete “must”
Done.
L297: Delete “unequivocally”
Done.
L310: Revise: “outcompetes” (overcomes?)
Done.
L311: Change “gaining ground” for “ spreading”
Done.
L317: Are there any relevant results for the comparator lines which should be mentioned?
Done.
L318: Change “Quadrable” to “Quadruple”
Done.
L329: Pls. revise to:
… the virus will spread through the vascular system of the infected plants …
Done.
L341: Revise expression: “…DNA virus of begomovirus.”
Done.
L342: Change to: “…viruses affecting tomato yield and causing abnormal leaf morphology, stunted growth, curled margins, and yellowing [9].”
Done.
L358: Reconsider heading: other headings indicate targeted virus
Done.
L372: Delete “specific”
Done.
L377: Revise the sentence
Done.
L378: Delete “set out to” and change examine to “examined”
Done.
L409-418: This description seems to be too technical and detailed.
Done.
L443: Delete “very well” and “matured and”
Done.
L446-448: Explain why heterozygous and homozygous plants have similar phenotypes.
Done.
L460-462: Delete “subsequently” and change to “ for modern large-scale crop production.”
Done.
L382. Pls. explain or revise:“commercially significant RNA viruses”?
Done.
L484-486: Change to:
Out of these plants 71.4% were edited at the Pro186 (CCA) codon, 98.7 % of them were modified at the C-14 position. Of these C-14 positions, 80% were C to T changes, 16.7% being homozygous mutants.
Done.
Conclusions
L490: Change to: “specific genes that render tomato plants susceptible to pests and pathogens..”
Done.
L505: revise to “in de-novo domestication” – Consider comments (5) above for revision!
Done.
L509: Revise “toomato”
Done.
Literaturverzeichnis
Agapito-Tenfen, Sarah Z.; Okoli, Arinze S.; Bernstein, Michael J.; Wikmark, Odd-Gunnar; Myhr, Anne I. (2018): Revisiting Risk Governance of GM Plants: The Need to Consider New and Emerging Gene-Editing Techniques. In: Frontiers in plant science 9, S. 1874. DOI: 10.3389/fpls.2018.01874.
Amoroso, Ciro Gianmaria; Panthee, Dilip R.; Andolfo, Giuseppe; Ramìrez, Felipe Palau; Ercolano, Maria Raffaella (2022): Genomic Tools for Improving Tomato to Biotic Stress Resistance. In: Chittaranjan Kole (Hg.): Genomic Designing for Biotic Stress Resistant Vegetable Crops. 1st ed. 2022. Cham: Springer International Publishing; Springer, S. 1–35. Online verfügbar unter https://link.springer.com/chapter/10.1007/978-3-030-97785-6_1.
Chandrasekaran, Murugesan; Boopathi, Thangavelu; Paramasivan, Manivannan (2021): A status-quo review on CRISPR-Cas9 gene editing applications in tomato. In: International Journal of Biological Macromolecules 190, S. 120–129. DOI: 10.1016/j.ijbiomac.2021.08.169.
Chu, Philomena; Agapito-Tenfen, Sarah Zanon (2022): Unintended Genomic Outcomes in Current and Next Generation GM Techniques: A Systematic Review. In: Plants (Basel, Switzerland) 11 (21), S. 2997. DOI: 10.3390/plants11212997.
Duensing, Nina; Sprink, Thorben; Parrott, Wayne A.; Fedorova, Maria; Lema, Martin A.; Wolt, Jeffrey D.; Bartsch, Detlef (2018): Novel Features and Considerations for ERA and Regulation of Crops Produced by Genome Editing. In: Frontiers in bioengineering and biotechnology 6, S. 79. DOI: 10.3389/fbioe.2018.00079.
Eckerstorfer, Michael F.; Dolezel, Marion; Engelhard, Margret; Giovannelli, Valeria; Grabowski, Marcin; Heissenberger, Andreas et al. (2023): Recommendations for the Assessment of Potential Environmental Effects of Genome-Editing Applications in Plants in the EU. In: Plants (Basel, Switzerland) 12 (9). DOI: 10.3390/plants12091764.
Kawall, Katharina; Cotter, Janet; Then, Christoph (2020): Broadening the GMO risk assessment in the EU for genome editing technologies in agriculture. In: Environ Sci Eur 32 (1). DOI: 10.1186/s12302-020-00361-2.
Kumar, S. Anil; Kottam, Suman Kumar; Narasu, M. Laxmi; Kumari, P. Hima (2022): Recent Trends in Targeting Genome Editing of Tomato for Abiotic and Biotic Stress Tolerance. In: S. H. Wani und G. Hensel (Hg.): Genome editing : current technology advances and applications for crop improvement: Springer, S. 273–285. Online verfügbar unter https://link.springer.com/chapter/10.1007/978-3-031-08072-2_15.
Li, Tingdong; Yang, Xinping; Yu, Yuan; Si, Xiaomin; Zhai, Xiawan; Zhang, Huawei et al. (2018): Domestication of wild tomato is accelerated by genome editing. In: Nat Biotechnol. DOI: 10.1038/nbt.4273.
Li, Yangyang; Wu, Xiuzhe; Zhang, Yan; Zhang, Qiang (2022): CRISPR/Cas genome editing improves abiotic and biotic stress tolerance of crops. In: Frontiers in genome editing 4, S. 987817. DOI: 10.3389/fgeed.2022.987817.
Salava, Hymavathi; Thula, Sravankumar; Mohan, Vijee; Kumar, Rahul; Maghuly, Fatemeh (2021): Application of Genome Editing in Tomato Breeding: Mechanisms, Advances, and Prospects. In: International Journal of Molecular Sciences 22 (2), S. 682. DOI: 10.3390/ijms22020682.
Zsögön, Agustin; ÄŒermák, Tomáš; Naves, Emmanuel Rezende; Notini, Marcela Morato; Edel, Kai H.; Weinl, Stefan et al. (2018): De novo domestication of wild tomato using genome editing. In: Nat Biotechnol. DOI: 10.1038/nbt.4272.

Reviewer 2 Report
Comments and Suggestions for Authors
I read the article carefully. I believe that the data presented in this review has some significance. However, it would be nice to see more work on the subject being studied: biotic stress-related gene editing in tomato. So far, not many target genes associated with tomato biotic stress have been edited using CRISPR-Cas, but not all known genes are included in this review.
1 There are other publications that describe knockout of genes associated with biotic stress, for example: - knockout of the SlBES1 gene, associated with participation in brassinosteroid signaling; the SlWAT1 gene, associated with increased resistance to Verticillium (Verticillium dahlia) and Fusarium wilt (Fusarium oxysporum); - knockout of DMR6, associated with susceptibility to bacterial and fungal diseases and others (Liu H. et all, 2021 https: //doi.org/10.1016/j.isci.2021.102926 Hanika K. et al., 2021 https:/ /doi.org/10.3389/fpls.2021.721674; Tomasella et al., 2021. https://doi.org/10.1073 /pnas.2026152118).
2 For a more complete review, it is necessary to present an analysis of works using of the CRISPR-Cas method for editing the tomato genome, noting methods for delivering the editing construct, a comprehensive assessment of the resulting mutant plants, and multiplex editing various genes and study the possible pleiotropic effect of target genes.
3 Gene names and species names should be italicized both in the manuscript and in the reference list. - Need corrected
Comments on the Quality of English LanguageThe English grammar, style and syntax used in the manuscript need some revision.
Author Response
This manuscript describes the use of the CRISPR/Cas9 genome editing system to improve resistance to biotic stress in tomatoes. While the topic of enhancing biotic stress resistance in tomatoes using CRISPR/Cas9 is undoubtedly important, the manuscript suffers from numerous critical flaws. These include significant issues with citation accuracy, redundancy, lack of novel insights, and poor organization. Technical aspects of implementing CRISPR/Cas9 in tomatoes are inadequately addressed, and the provided references are often irrelevant or incorrect. Moreover, several sections are repetitive or unnecessarily fragmented, detracting from the coherence and impact of the review. Given these substantial shortcomings, I recommend that the manuscript be rejected for publication.
We are appreciating the reviewer recommendation and his efforts in providing key points to enhance the level of the manuscript. We therefore worked on the manuscript to afford the reviewer comments and suggestions.
However, all the authors hereby confirm that the manuscript was modified based on the comments and suggestions listed by the reviewer.
Introduction
Line 30: The letter L. (Linnaeus) in the species name should be written without italics.
Modified.
Lines 53-55: Check your sentence structure. It looks like the point was made incorrectly.
Modified.
Line 62: The expression “lack of ability to modify monocot crops” is incorrect. This natural limitation was solved long ago using special strains. The links provided to support this phrase do not correspond to the context.
Modified, and references was changed accordingly.
Lines 68-69: Should be rephrased to avoid duplication of the word “tomato”.
Done.
Line 70: Remove the word "tomato." It is clear from the context what plant is being discussed.
Done.
Line 77: Re-check the structure of the sentence.
Modified
Line 80: This is certainly not true. There are several fairly recent reviews in the literature devoted specifically to increasing the biotic resistance of tomatoes using the CRISPR/Cas9 method.
Modified. And some of these publications were added. We modified that our manuscript was predominantly focused on the biotic stress only.
A note on the entire first chapter: The authors provide links that are completely inappropriate in context. For example, for the phrase “Biotic stress not only results in significantly reduced crop quality and yield, but also changes plant physiology, decrease biomass, decrease seed set, accumulate protective metabolites” there is an article with the title “Agrobacterium-mediated plant transformation: biology and applications.” The phrase “These programs include chemical or irradiation mutagenesis, translocation breeding, and intergeneric crosses” is supported by the article “Plant pattern recognition receptor complexes at the plasma membrane.” These are not isolated examples—such incorrect links are prevalent throughout the text. This is a very serious mistake, which indicates the inability of the authors to work with sources. For a review, this is inexcusable.
Thanks for withdrawing our attention to this serious mistake which occurred during transferring our initial version to the template of the MDPI-plants.
Chapter 2
There are no references to Figure 1 or Table 1 in the text. Additionally, the contents of Table 1 are not discussed in any way. Simply listing information about different types of nucleases without context or discussion is meaningless. Furthermore, the table signature must be capitalized.
Figure 1 and Table 1 were cited. Table 1 was discussed in the text.
Line 118: "Tomatoes" should begin with a small letter.
Modified.
Line 121: The authors describe the use of the CRISPR/Cas9 system on tomatoes as easy and effective. However, they do not address the technical aspects of implementing the CRISPR/Cas9 genome editing system specifically in tomatoes. Details on how its components are introduced, how regenerated plants are obtained, how to eliminate transgenic elements and obtain transgene-free mutant lines, and how mutant events and off-target effects are verified are missing.
The requested part with all information was added to section 2.
Line 122: The word “tool” is misspelled, and there is a link to a figure without parentheses.
Modified
In Figure 2: The word "weed" is not capitalized. Additionally, classifying the effects of herbicides as biotic stress is incorrect.
Modified. We provided herbicides resistance as a tool for supporting weed treatment
Line 129: "Article" should begin with a small letter.
Line 131: The species name should not be capitalized, and the entire name should be in italics.
Modified
Lines 148 and 149: The names of the mutant plants are inconsistent.
Modified
Line 161: Rephrase to "the pathogen then upregulates the gene for DNA-binding protein from starved cells (Dps)." The gene name should be in italics.
This part was deleted.
Line 162: The Latin name for bacteria should be in italics. The first mention of E. coli should include the full genus name.
This part was deleted.
Lines 169-175: This last paragraph does not relate to resistance to bacterial pathogens.
This part was deleted.
Line 179: "L. Kiss" should not be in italics.
Modified.
Line 191: The gene structure of the Mlo loci was established by different authors. The provided link is not relevant. Similarly, reference 9 on Line 195 is not an original study but a review. Much research has focused on the Mlo genes in tomatoes, and this chapter needs to be completely rewritten to emphasize original research.
The whole part was restructured and completely rewritten and the references were adopted accordingly.
Line 197: "Powdery" should begin with a small letter.
Modified.
From Chapter 3.1 onward: The authors use a flawed technique in their descriptions. They take one or two articles on editing a trait in tomatoes and use them to write entire chapters, including introductory parts that describe the theoretical background. Additionally, in many cases (e.g., Chapter 3.2.2., 3.4.2), the authors provide excessive details on works not related to the review's subject. Essentially, these individual chapters are descriptions of cited works without citations of original research.
Thanks for withdrawing our attention to this point, we modified the manuscript so that the parts become more advanced and include several references. Moreover, Chapters 3.2.2 and 3.4.2 were modified not to include excessive unrelated details and extra studies were added. The whole part was restructured according to the reviewer suggestion.
Chapters 3.3.4 and 3.3.5 are devoted to the same mechanism of participation of DCL and microRNA in resistance to viruses. Moreover, the same tobacco mosaic virus is studied, referred to as ToMV in the first case and TMV in the second. The authors separate these discussions based on cited articles, although it would be more logical to combine these chapters. Additionally, the introductory part of Chapter 3.3.5 is written as if the role of viruses and microRNAs in protection against them had not already been discussed earlier (in 3.3.4.).
3.3.4 and 3.3.5 parts were combines and were presented in a logical manner as requested. Started with the tomato mosaic virus (ToMV) study and then the other study of tobacco mosaic virus (TMV). The introductory sentences were normalized for both of them.
Line 426. This is Figure 3, not Figure 1. The captions are very small and difficult to distinguish. Additionally, the authors claim that this figure “shows that plant hormones like strigolactones interact with seeds to enable germination,” but this is not the case.
The caption was modified regarding the seed germination. However the font size is settled by the MDPI template.
Conclusion
Lines 496-497. The authors do not describe the modification of tomato “without introducing foreign DNA”. On the contrary, all the cited sources refer to genetic transformation.
Thanks for withdrawing our attention to this point. We illustrated this point in section 2 and illustrated how the segregation can further illuminate the foreign DNA.
Lines 499-501. None of the listed limitations “off-target effects, cost, complexity, and potential future consequences” were mentioned in the work, so it is not appropriate to talk about them in the conclusion.
Thanks for withdrawing our attention to this point. We provided some extra information regarding off-target in different parts of the text.
Lines 503-506. These are particulars that were not described in the work and are not appropriate in the conclusion.
We added part 4 which discusses multiplex genome editing in tomato.
Lines 507-509. What are these editing tools? They are first mentioned only in the conclusion.
Yes, we understand that the manuscript focuses on CRISPR/Cas9. We added these tools as promising editing tools to be used for future research advances for editing biotic stress related genes in tomato.
In general, the conclusion of the manuscript does not correspond to the content presented. It fails to provide a cohesive summary of the findings and their implications for future research and practical applications. The lack of a strong, accurate, and insightful conclusion further weakens the overall impact of the manuscript.
We advanced the conclusion as requested.

Reviewer 3 Report
Comments and Suggestions for Authors
This manuscript describes the use of the CRISPR/Cas9 genome editing system to improve resistance to biotic stress in tomatoes. While the topic of enhancing biotic stress resistance in tomatoes using CRISPR/Cas9 is undoubtedly important, the manuscript suffers from numerous critical flaws. These include significant issues with citation accuracy, redundancy, lack of novel insights, and poor organization. Technical aspects of implementing CRISPR/Cas9 in tomatoes are inadequately addressed, and the provided references are often irrelevant or incorrect. Moreover, several sections are repetitive or unnecessarily fragmented, detracting from the coherence and impact of the review. Given these substantial shortcomings, I recommend that the manuscript be rejected for publication.
Introduction
Line 30: The letter L. (Linnaeus) in the species name should be written without italics.
Lines 53-55: Check your sentence structure. It looks like the point was made incorrectly.
Line 62: The expression “lack of ability to modify monocot crops” is incorrect. This natural limitation was solved long ago using special strains. The links provided to support this phrase do not correspond to the context.
Lines 68-69: Should be rephrased to avoid duplication of the word “tomato”.
Line 70: Remove the word "tomato." It is clear from the context what plant is being discussed.
Line 77: Re-check the structure of the sentence.
Line 80: This is certainly not true. There are several fairly recent reviews in the literature devoted specifically to increasing the biotic resistance of tomatoes using the CRISPR/Cas9 method.
A note on the entire first chapter: The authors provide links that are completely inappropriate in context. For example, for the phrase “Biotic stress not only results in significantly reduced crop quality and yield, but also changes plant physiology, decrease biomass, decrease seed set, accumulate protective metabolites” there is an article with the title “Agrobacterium-mediated plant transformation: biology and applications.” The phrase “These programs include chemical or irradiation mutagenesis, translocation breeding, and intergeneric crosses” is supported by the article “Plant pattern recognition receptor complexes at the plasma membrane.” These are not isolated examples—such incorrect links are prevalent throughout the text. This is a very serious mistake, which indicates the inability of the authors to work with sources. For a review, this is inexcusable.
Chapter 2
There are no references to Figure 1 or Table 1 in the text. Additionally, the contents of Table 1 are not discussed in any way. Simply listing information about different types of nucleases without context or discussion is meaningless. Furthermore, the table signature must be capitalized.
Line 118: "Tomatoes" should begin with a small letter.
Line 121: The authors describe the use of the CRISPR/Cas9 system on tomatoes as easy and effective. However, they do not address the technical aspects of implementing the CRISPR/Cas9 genome editing system specifically in tomatoes. Details on how its components are introduced, how regenerated plants are obtained, how to eliminate transgenic elements and obtain transgene-free mutant lines, and how mutant events and off-target effects are verified are missing.
Line 122: The word “tool” is misspelled, and there is a link to a figure without parentheses.
In Figure 2: The word "weed" is not capitalized. Additionally, classifying the effects of herbicides as biotic stress is incorrect.
Line 129: "Article" should begin with a small letter.
Line 131: The species name should not be capitalized, and the entire name should be in italics.
Lines 148 and 149: The names of the mutant plants are inconsistent.
Line 161: Rephrase to "the pathogen then upregulates the gene for DNA-binding protein from starved cells (Dps)." The gene name should be in italics.
Line 162: The Latin name for bacteria should be in italics. The first mention of E. coli should include the full genus name.
Lines 169-175: This last paragraph does not relate to resistance to bacterial pathogens.
Line 179: "L. Kiss" should not be in italics.
Line 191: The gene structure of the Mlo loci was established by different authors. The provided link is not relevant. Similarly, reference 9 on Line 195 is not an original study but a review. Much research has focused on the Mlo genes in tomatoes, and this chapter needs to be completely rewritten to emphasize original research.
Line 197: "Powdery" should begin with a small letter.
From Chapter 3.1 onward: The authors use a flawed technique in their descriptions. They take one or two articles on editing a trait in tomatoes and use them to write entire chapters, including introductory parts that describe the theoretical background. Additionally, in many cases (e.g., Chapter 3.2.2., 3.4.2), the authors provide excessive details on works not related to the review's subject. Essentially, these individual chapters are descriptions of cited works without citations of original research.
Chapters 3.3.4 and 3.3.5 are devoted to the same mechanism of participation of DCL and microRNA in resistance to viruses. Moreover, the same tobacco mosaic virus is studied, referred to as ToMV in the first case and TMV in the second. The authors separate these discussions based on cited articles, although it would be more logical to combine these chapters. Additionally, the introductory part of Chapter 3.3.5 is written as if the role of viruses and microRNAs in protection against them had not already been discussed earlier (in 3.3.4.).
Line 426. This is Figure 3, not Figure 1. The captions are very small and difficult to distinguish. Additionally, the authors claim that this figure “shows that plant hormones like strigolactones interact with seeds to enable germination,” but this is not the case.
Conclusion
Lines 496-497. The authors do not describe the modification of tomato “without introducing foreign DNA”. On the contrary, all the cited sources refer to genetic transformation.
Lines 499-501. None of the listed limitations “off-target effects, cost, complexity, and potential future consequences” were mentioned in the work, so it is not appropriate to talk about them in the conclusion.
Lines 503-506. These are particulars that were not described in the work and are not appropriate in the conclusion.
Lines 507-509. What are these editing tools? They are first mentioned only in the conclusion.
In general, the conclusion of the manuscript does not correspond to the content presented. It fails to provide a cohesive summary of the findings and their implications for future research and practical applications. The lack of a strong, accurate, and insightful conclusion further weakens the overall impact of the manuscript.
Author Response
I read the article carefully. I believe that the data presented in this review has some significance. However, it would be nice to see more work on the subject being studied: biotic stress-related gene editing in tomato. So far, not many target genes associated with tomato biotic stress have been edited using CRISPR-Cas, but not all known genes are included in this review.
We would like to thank the reviewer for the valuable comments and suggestions. We worked on all the suggestions thoroughly and all authors confirm that the requested modifications were done where are needed.
1 There are other publications that describe knockout of genes associated with biotic stress, for example: - knockout of the SlBES1 gene, associated with participation in brassinosteroid signaling; the SlWAT1 gene, associated with increased resistance to Verticillium (Verticillium dahlia) and Fusarium wilt (Fusarium oxysporum); - knockout of DMR6, associated with susceptibility to bacterial and fungal diseases and others (Liu H. et all, 2021 https: //doi.org/10.1016/j.isci.2021.102926 Hanika K. et al., 2021 https:/ /doi.org/10.3389/fpls.2021.721674; Tomasella et al., 2021. https://doi.org/10.1073 /pnas.2026152118).
Done . – the requested studies were added where needed.
2 For a more complete review, it is necessary to present an analysis of works using of the CRISPR-Cas method for editing the tomato genome, noting methods for delivering the editing construct, a comprehensive assessment of the resulting mutant plants, and multiplex editing various genes and study the possible pleiotropic effect of target genes.
Done . – The requested parts were added in sections 2 and 4.
3 Gene names and species names should be italicized both in the manuscript and in the reference list. - Need corrected
Done . – Gene names and speces names were italicized.

Round 2
Reviewer 1 Report
Comments and Suggestions for Authors
Dear authors
Thanks for the revision which in part introduced substantial improvements. However, not all shortcomings and comments were addressed sufficiently and some of the revisions introduced further flaws and unfortunately even factual errors. In the following some of the issues are mentioned. In summary another round of revision seems to be necessary to get the manuscript into a shape that is sufficient for publication.
1) Editorial errors
While many of the previously noted editing issues have been addressed, further editorial mistakes have been introduced throughout the whole manuscript by adding new text: These flaws range from awkward expressions (e.g. L92 start of sentence), grammatical errors (e.g. L99: application; L164: specificity - singular instead of plural), misspellings (L788: Theese), missing verbs (L793), abbreviations that are not introduced (e.g. L127 DSB), references that are not added in MDPI Plants style (e.g. L149, L7785ff), as well as text that is not formatted in MDPI Plants style (L811ff).
Another round of thorough editing to correct suchmistakes throughout the whole manuscript is needed! Mind that the above list is not an exhaustive list of all errors, but merely a collection of a few different examples.
2) References
As stated under (1) many references have been added, however a substantial number of these not according to the style of the template. This is particularly apparent in the newly added chapter 4; but also in other parts of the manuscript. The wrongly formatted references are also not added to the reference list and thus cannot be checked for appropriateness.
According to the responses by the authors the previously suggested references have been added – this seems not to be the case without providing explanations. Non-considerations of this literature is reflected in factual errors as indicated below.
3) Content
Evaluation of content revision is also considered ambigous: On the one hand a lot of new details have been added and some parts of the manuscript were improved structurally. On the other hand some revisions missed to address the issues raised with the initial comments – e.g. previous comment (5) which was only partly addressed and the revision misses the main issue that De Novo Domestication is based on an approach to develop plants with increased resistance to biotic and abiotic stressors, that is fundamentally different to the editing of plant genes that are directly responsible for stress resistance (as discussed in L811-825). That this is based on multiplex editing is a different aspect and the technical review of multiplex editing doesn´t capture the uniqueness of the De Novo Domestication approach. Thus the revised text doesn´t sufficiently address the comment or provide references to that effect!
In addition the comment to revise the conclusion section was not addressed properly and new factual errors have been introduced with the minor revisions that were made to the end of this section (L871-874): The statement that genome editing methods are prohibited in the EU, Japan and China is plainly wrong and urgently needs to be revised! Japan has recently introduced specific legislation to address genome edited organisms and in 2021 cleared some genome edited plants and animals for commercial sale, e.g. genome-edited, growth –enhanced Red Sea bream and tiger puff and genome edited tomato variety with an increased content of gamma-aminobutyric acid (GABA tomato). Chinese scientists are leading developers of genome edited organisms as is apparent from the record of scientific publications in the area (see e.g. EU-Sage-database) and in the EU genome edited organisms arenot prohibited, but currently regulated in a similar manner as GMOs (authorisation is granted after a premarket-risk assessment, labelling and post market monitoring are required during commercial use). This is comprehensively described in Eckerstorfer et al. 2023 Plants 2023, 12, 1764. https://doi.org/10.3390/plants12091764 (see chpt 1 Introduction)!
1) While many of the previously noted editing issues have been addressed, further editorial mistakes have been introduced throughout the whole manuscript by adding new text: These flaws range from awkward expressions (e.g. L92 start of sentence), grammatical errors (e.g. L99: application; L164: specificity - singular instead of plural), misspellings (L788: Theese), missing verbs (L793), abbreviations that are not introduced (e.g. L127 DSB), references that are not added in MDPI Plants style (e.g. L149, L7785ff), as well as text that is not formatted in MDPI Plants style (L811ff).
Another round of thorough editing to correct suchmistakes throughout the whole manuscript is needed! Mind that the above list is not an exhaustive list of all errors, but merely a collection of a few different examples.
Author Response
Dear authors
Thanks for the revision which in part introduced substantial improvements. However, not all shortcomings and comments were addressed sufficiently and some of the revisions introduced further flaws and unfortunately even factual errors. In the following some of the issues are mentioned. In summary another round of revision seems to be necessary to get the manuscript into a shape that is sufficient for publication.
We would like to extend our sincere thanks for your valuable suggestions and comments on our manuscript. We believe that your insights have significantly refined and improved the quality of our work. Your careful review and constructive feedback have been instrumental in fine-tuning the manuscript.
Thank you once again for your time and effort in reviewing our submission. We greatly appreciate your contribution to our research.
1) Editorial errors
While many of the previously noted editing issues have been addressed, further editorial mistakes have been introduced throughout the whole manuscript by adding new text: These flaws range from awkward expressions (e.g. L92 start of sentence), grammatical errors (e.g. L99: application; L164: specificity - singular instead of plural), misspellings (L788: Theese), missing verbs (L793), abbreviations that are not introduced (e.g. L127 DSB), references that are not added in MDPI Plants style (e.g. L149, L7785ff), as well as text that is not formatted in MDPI Plants style (L811ff).
Another round of thorough editing to correct suchmistakes throughout the whole manuscript is needed! Mind that the above list is not an exhaustive list of all errors, but merely a collection of a few different examples.
We conducted all the requested modifications such as:
L92 start of sentence: Done, Line 78. L99: application: Done, Line 84. L164: specificity - singular instead of plural. Done, Line 148. L788: Theese. Modified, Line 610. L793: missing verbs. Done, Line 613. L811: MDPI format. Done, Line 629.
Further editing round was conducted for the whole manuscript to correct errors, references, and MDPI style.
2) References
As stated under (1) many references have been added, however a substantial number of these not according to the style of the template. This is particularly apparent in the newly added chapter 4; but also in other parts of the manuscript. The wrongly formatted references are also not added to the reference list and thus cannot be checked for appropriateness.
According to the responses by the authors the previously suggested references have been added – this seems not to be the case without providing explanations. Non-considerations of this literature is reflected in factual errors as indicated below.
We revised the whole manuscript, modified all references into the MDPI format, and confirmed their addition to the reference list.
3) Content
Evaluation of content revision is also considered ambigous: On the one hand a lot of new details have been added and some parts of the manuscript were improved structurally. On the other hand some revisions missed to address the issues raised with the initial comments – e.g. previous comment (5) which was only partly addressed and the revision misses the main issue that De Novo Domestication is based on an approach to develop plants with increased resistance to biotic and abiotic stressors, that is fundamentally different to the editing of plant genes that are directly responsible for stress resistance (as discussed in L811-825). That this is based on multiplex editing is a different aspect and the technical review of multiplex editing doesn´t capture the uniqueness of the De Novo Domestication approach. Thus the revised text doesn´t sufficiently address the comment or provide references to that effect!
We modified the text as suggested, please check the new version line 601-612.
In addition the comment to revise the conclusion section was not addressed properly and new factual errors have been introduced with the minor revisions that were made to the end of this section (L871-874): The statement that genome editing methods are prohibited in the EU, Japan and China is plainly wrong and urgently needs to be revised! Japan has recently introduced specific legislation to address genome edited organisms and in 2021 cleared some genome edited plants and animals for commercial sale, e.g. genome-edited, growth –enhanced Red Sea bream and tiger puff and genome edited tomato variety with an increased content of gamma-aminobutyric acid (GABA tomato). Chinese scientists are leading developers of genome edited organisms as is apparent from the record of scientific publications in the area (see e.g. EU-Sage-database) and in the EU genome edited organisms arenot prohibited, but currently regulated in a similar manner as GMOs (authorisation is granted after a premarket-risk assessment, labelling and post market monitoring are required during commercial use). This is comprehensively described in Eckerstorfer et al. 2023 Plants 2023, 12, 1764. https://doi.org/10.3390/plants12091764 (see chpt 1 Introduction)!
We modified the text as recommended, please check the new version line 601-612
Comments on the Quality of English Language
1) While many of the previously noted editing issues have been addressed, further editorial mistakes have been introduced throughout the whole manuscript by adding new text: These flaws range from awkward expressions (e.g. L92 start of sentence), grammatical errors (e.g. L99: application; L164: specificity - singular instead of plural), misspellings (L788: Theese), missing verbs (L793), abbreviations that are not introduced (e.g. L127 DSB), references that are not added in MDPI Plants style (e.g. L149, L7785ff), as well as text that is not formatted in MDPI Plants style (L811ff).
Another round of thorough editing to correct suchmistakes throughout the whole manuscript is needed! Mind that the above list is not an exhaustive list of all errors, but merely a collection of a few different examples.
Thanks for withdrawing our attention to these points. We conducted all the requested modifications such as:
L92 start of sentence: Done, Line 78. L99: application: Done, Line 84. L164: specificity - singular instead of plural. Done, Line 148. L788: Theese. Modified, Line 610. L793: missing verbs. Done, Line 613. L811: MDPI format. Done, Line 629.
Further editing round was conducted for the whole manuscript to correct errors, references and MDPI style.
Reviewer 2 Report
Comments and Suggestions for Authors
Check the text for italics in gene names
Author Response
Check the text for italics in gene names.
Thanks so much for withdrawing our attention to this point, we revised the manuscript for the gene names and italicized it.
Reviewer 3 Report
Comments and Suggestions for Authors
In the revised manuscript, the authors have done significant work to improve the work. However, the authors' attached response is addressed to another Reviewer. This is a clear mistake.
I will refrain from making a final response on this manuscript until I receive a detailed response specifically to my first review.
Author Response
In the revised manuscript, the authors have done significant work to improve the work. However, the authors' attached response is addressed to another Reviewer. This is a clear mistake.
I will refrain from making a final response on this manuscript until I receive a detailed response specifically to my first review.
We apologize for the oversight in our previous response. It seems there was a mix-up, and we inadvertently attached the response intended for another reviewer. We sincerely appreciate your patience and the valuable feedback you have provided.
Below are our detailed responses to the points raised in your initial review. We have carefully addressed all your comments and made the necessary revisions to the manuscript.
Thank you for your understanding and for allowing us to correct this mistake. We look forward to your feedback on the revised manuscript.
This manuscript describes the use of the CRISPR/Cas9 genome editing system to improve resistance to biotic stress in tomatoes. While the topic of enhancing biotic stress resistance in tomatoes using CRISPR/Cas9 is undoubtedly important, the manuscript suffers from numerous critical flaws. These include significant issues with citation accuracy, redundancy, lack of novel insights, and poor organization. Technical aspects of implementing CRISPR/Cas9 in tomatoes are inadequately addressed, and the provided references are often irrelevant or incorrect. Moreover, several sections are repetitive or unnecessarily fragmented, detracting from the coherence and impact of the review. Given these substantial shortcomings, I recommend that the manuscript be rejected for publication.
We are appreciating the reviewer's recommendation and his efforts in providing key points to enhance the level of the manuscript. We therefore worked on the manuscript to afford the reviewer's comments and suggestions.
However, all the authors hereby confirm that the manuscript was modified based on the comments and suggestions listed by the reviewer.
Introduction
Line 30: The letter L. (Linnaeus) in the species name should be written without italics.
Modified.
Lines 53-55: Check your sentence structure. It looks like the point was made incorrectly.
Modified.
Line 62: The expression “lack of ability to modify monocot crops” is incorrect. This natural limitation was solved long ago using special strains. The links provided to support this phrase do not correspond to the context.
Modified, and references were changed accordingly.
Lines 68-69: Should be rephrased to avoid duplication of the word “tomato”.
Done.
Line 70: Remove the word "tomato." It is clear from the context what plant is being discussed.
Done.
Line 77: Re-check the structure of the sentence.
Modified
Line 80: This is certainly not true. There are several fairly recent reviews in the literature devoted specifically to increasing the biotic resistance of tomatoes using the CRISPR/Cas9 method.
Modified. Some of these publications were added. We modified that our manuscript was predominantly focused on the biotic stress only.
A note on the entire first chapter: The authors provide links that are completely inappropriate in context. For example, for the phrase “Biotic stress not only results in significantly reduced crop quality and yield, but also changes plant physiology, decrease biomass, decrease seed set, accumulate protective metabolites” there is an article with the title “Agrobacterium-mediated plant transformation: biology and applications.” The phrase “These programs include chemical or irradiation mutagenesis, translocation breeding, and intergeneric crosses” is supported by the article “Plant pattern recognition receptor complexes at the plasma membrane.” These are not isolated examples—such incorrect links are prevalent throughout the text. This is a very serious mistake, which indicates the inability of the authors to work with sources. For a review, this is inexcusable.
Thanks for withdrawing our attention to this serious mistake that occurred during the transfer of our initial version to the template of the MDPI-plants.
Chapter 2
There are no references to Figure 1 or Table 1 in the text. Additionally, the contents of Table 1 are not discussed in any way. Simply listing information about different types of nucleases without context or discussion is meaningless. Furthermore, the table signature must be capitalized.
Figure 1 and Table 1 were cited. Table 1 was discussed in the text.
Line 118: "Tomatoes" should begin with a small letter.
Modified.
Line 121: The authors describe the use of the CRISPR/Cas9 system on tomatoes as easy and effective. However, they do not address the technical aspects of implementing the CRISPR/Cas9 genome editing system specifically in tomatoes. Details on how its components are introduced, how regenerated plants are obtained, how to eliminate transgenic elements and obtain transgene-free mutant lines, and how mutant events and off-target effects are verified are missing.
The requested part with all information was added to section 2.
Line 122: The word “tool” is misspelled, and there is a link to a figure without parentheses.
Modified
In Figure 2: The word "weed" is not capitalized. Additionally, classifying the effects of herbicides as biotic stress is incorrect.
Modified. We provided herbicides resistance as a tool for supporting weed treatment
Line 129: "Article" should begin with a small letter.
Modified
Line 131: The species name should not be capitalized, and the entire name should be in italics.
Modified
Lines 148 and 149: The names of the mutant plants are inconsistent.
Modified
Line 161: Rephrase to "the pathogen then upregulates the gene for DNA-binding protein from starved cells (Dps)." The gene name should be in italics.
This part was deleted.
Line 162: The Latin name for bacteria should be in italics. The first mention of E. coli should include the full genus name.
This part was deleted based on another reviewer's recommendation.
Lines 169-175: This last paragraph does not relate to resistance to bacterial pathogens.
This part was deleted.
Line 179: "L. Kiss" should not be in italics.
Modified.
Line 191: The gene structure of the Mlo loci was established by different authors. The provided link is not relevant. Similarly, reference 9 on Line 195 is not an original study but a review. Much research has focused on the Mlo genes in tomatoes, and this chapter needs to be completely rewritten to emphasize original research.
The whole part was restructured and completely rewritten and the references were adopted accordingly.
Line 197: "Powdery" should begin with a small letter.
Modified.
From Chapter 3.1 onward: The authors use a flawed technique in their descriptions. They take one or two articles on editing a trait in tomatoes and use them to write entire chapters, including introductory parts that describe the theoretical background. Additionally, in many cases (e.g., Chapter 3.2.2., 3.4.2), the authors provide excessive details on works not related to the review's subject. Essentially, these individual chapters are descriptions of cited works without citations of original research.
Thanks for withdrawing our attention to this point, we modified the manuscript so that the parts become more advanced and include several references. Moreover, Chapters 3.2.2 and 3.4.2 were modified not to include excessive unrelated details, and extra studies were added. The whole part was restructured according to the reviewer's suggestion.
Chapters 3.3.4 and 3.3.5 are devoted to the same mechanism of participation of DCL and microRNA in resistance to viruses. Moreover, the same tobacco mosaic virus is studied, referred to as ToMV in the first case and TMV in the second. The authors separate these discussions based on cited articles, although it would be more logical to combine these chapters. Additionally, the introductory part of Chapter 3.3.5 is written as if the role of viruses and microRNAs in protection against them had not already been discussed earlier (in 3.3.4.).
3.3.4 and 3.3.5 parts were combines and were presented in a logical manner as requested. Started with the tomato mosaic virus (ToMV) study and then the other study of tobacco mosaic virus (TMV). The introductory sentences were normalized for both of them.
Line 426. This is Figure 3, not Figure 1. The captions are very small and difficult to distinguish. Additionally, the authors claim that this figure “shows that plant hormones like strigolactones interact with seeds to enable germination,” but this is not the case.
The caption was modified regarding the seed germination. However, the font size is settled by the MDPI template.
Conclusion
Lines 496-497. The authors do not describe the modification of tomato “without introducing foreign DNA”. On the contrary, all the cited sources refer to genetic transformation.
Thanks for withdrawing our attention to this point. We illustrated this point in section 2 and illustrated how the segregation can further illuminate the foreign DNA.
Lines 499-501. None of the listed limitations “off-target effects, cost, complexity, and potential future consequences” were mentioned in the work, so it is not appropriate to talk about them in the conclusion.
Thanks for withdrawing our attention to this point. We provided some extra information regarding off-target in different parts of the text.
Lines 503-506. These are particulars that were not described in the work and are not appropriate in the conclusion.
We added part 4 which discusses multiplex genome editing in tomato.
Lines 507-509. What are these editing tools? They are first mentioned only in the conclusion.
Yes, we understand that the manuscript focuses on CRISPR/Cas9. We added these tools as promising editing tools to be used for future research advances for editing biotic stress-related genes in tomato.
In general, the conclusion of the manuscript does not correspond to the content presented. It fails to provide a cohesive summary of the findings and their implications for future research and practical applications. The lack of a strong, accurate, and insightful conclusion further weakens the overall impact of the manuscript.
We advanced the conclusion as requested.
Round 3
Reviewer 1 Report
Comments and Suggestions for Authors
Dear Authors
Kind thanks for addressing the submitted comments and for providing an additional round of revisions - this really improved consistency and readability in a way that I believe that the manuscript is good for publication with attention given to a few very minor things that can be corrected during final stages of production:
L223: Consider the expression "more durable"
L307: Change they to They (with capital letter at start of sentence)
L315: consider "lower" instead of "slowed"
L346: defeated sounds a little weird - maybe consider using "disabled" instead
L492: change to: "targeted by two RNA viruses..."
L576: consider moving up the abbreviation (EPSPS) to immediately after the abbreviated enzyme
L604: change "suit" for "suite"
L611: maybe use the expression: "parental wild plant species" (including "modified" might be confusing in that context)
L671: change to "is considered" and indicate who is considering this.
L681ff: Reconsider the expression "popularity" (maybe acceptability is better). I suggest based on my understanding of the sentences: "As public awareness is also necessary for the acceptability of genome-edited genotypes, which no longer maintain any foreign genes in their genetic background, governmental and societal concerns must continue to be addressed."
L686: delete "non-adopters"
Comments on the Quality of English Language
As indicated above the following minor changes are suggested to be included in final production:
L223: Consider the expression "more durable"
L307: Change they to They (with capital letter at start of sentence)
L315: consider "lower" instead of "slowed"
L346: defeated sounds a little weird - maybe consider using "disabled" instead
L492: change to: "argeted by two RNA viruses..."
L576: consider moving up the abbreviation (EPSPS) to immediately after the abbreviated enzyme
L604: change "suit" for "suite"
L611: maybe use the expression: "parental wild plant species" (including "modified" might be confusing in that context)
L671: change to "is considered" and indicate who is considering this.
L681ff: Reconsider the expression "popularity" (maybe acceptability is better). I suggest based on my understanding of the sentences: "As public awareness is also necessary for the acceptability of genome-edited genotypes, which no longer maintain any foreign genes in their genetic background, governmental and societal concerns must continue to be addressed."
L686: delete "non-adopters"
Author Response
Dear Authors
Kind thanks for addressing the submitted comments and for providing an additional round of revisions - this really improved consistency and readability in a way that I believe that the manuscript is good for publication with attention given to a few very minor things that can be corrected during final stages of production:
Thank you for your kind feedback and for recognizing the improvements made to our manuscript. We greatly appreciate your insightful comments and suggestions, which have undoubtedly enhanced the consistency and readability of our work.
We carefully addressed the minor corrections you indicated during the final stages of the review to ensure the highest quality of our manuscript.
Thank you once again for your thorough review and valuable contributions to our manuscript.
L223: Consider the expression "more durable"
Done line 227
L307: Change they to They (with capital letter at start of sentence)
Done
L315: consider "lower" instead of "slowed"
Done line 327
L346: defeated sounds a little weird - maybe consider using "disabled" instead
Done line 358
L492: change to: "targeted by two RNA viruses..."
Done line 505.
L576: consider moving up the abbreviation (EPSPS) to immediately after the abbreviated enzyme
Done line 586.
L604: change "suit" for "suite"
Done line 611.
L611: maybe use the expression: "parental wild plant species" (including "modified" might be confusing in that context)
Done line 618.
L671: change to "is considered" and indicate who is considering this.
Done line 684.
L681ff: Reconsider the expression "popularity" (maybe acceptability is better). I suggest based on my understanding of the sentences: "As public awareness is also necessary for the acceptability of genome-edited genotypes, which no longer maintain any foreign genes in their genetic background, governmental and societal concerns must continue to be addressed."
Many thanks for the suggestion, looks clearer and more defined. Done line 684.
L686: delete "non-adopters"
Done line 703.
Comments on the Quality of English Language
As indicated above the following minor changes are suggested to be included in final production:
L223: Consider the expression "more durable"
L307: Change they to They (with capital letter at start of sentence)
L315: consider "lower" instead of "slowed"
L346: defeated sounds a little weird - maybe consider using "disabled" instead
L492: change to: "argeted by two RNA viruses..."
L576: consider moving up the abbreviation (EPSPS) to immediately after the abbreviated enzyme
L604: change "suit" for "suite"
L611: maybe use the expression: "parental wild plant species" (including "modified" might be confusing in that context)
L671: change to "is considered" and indicate who is considering this.
L681ff: Reconsider the expression "popularity" (maybe acceptability is better). I suggest based on my understanding of the sentences: "As public awareness is also necessary for the acceptability of genome-edited genotypes, which no longer maintain any foreign genes in their genetic background, governmental and societal concerns must continue to be addressed."
L686: delete "non-adopters"
All the comments were modified accordingly.
Reviewer 3 Report
Comments and Suggestions for Authors
The revised manuscript has been significantly improved. However, a thorough review and correction of punctuation, formatting, and content structure are recommended to elevate the article to a higher standard of academic excellence.
Line 128-129: It seems that the reference to Agrobacterium-mediated transformation of tomato cotyledons is missing here.
In Figure 2: The word "weed" is not capitalized, while other examples (Fungi, Bacteria, Viruses, etc.) are written with a capital letter.
Line 179: Remove the Latin name of tomato, as it is redundant here.
Line 212: Both full and abbreviated names of proteins (PRP1 and DEA1) are written without italics. Replace the abbreviation "wt" with the full spelling "wild type."
Line 265: The word "gene" is written here without italics.
Line 270: The Latin name of the fungus is written in italics.
Line 274: The "pmr4" mutation and the Latin name of the fungus are written in italics.
References 71 and 75 are the same. Additionally, the last paragraph in Chapter 3.2.1 repeats the second paragraph. The chapter needs to be rewritten to avoid repetition and to cite correctly.
Line 290-293: This is the first mention of microRNA and therefore requires a full description. In addition, "miRNA" (short for microRNA) is written with a lowercase letter. The authors' statement here is not entirely correct. Mature tomato miR482b and miR482c have a specific length of 22 nucleotides, while in general miRNA can have a length of 20 to 24 nucleotides. The beginning of this paragraph needs to be rephrased for clarity.
Line 318-319: Protein names should be written without italics.
Line 324-339: There is a repetition of sentences. Fusarium wilt has been shortened to FW; use the abbreviation consistently.
Line 396: "B. cinerea" should be written in italics.
Line 405: A reference to the study is missing.
Line 412: "Tomato" should be written with a lowercase letter.
Line 416: "Arabidopsis" should be written in italics.
Line 417: "TOM1" should be written without brackets, and the extra period and bracket at the end of the sentence should be removed. If referring to a protein, "TOM1" should be written without italics.
Line 424: From here on, replace "wt" with "wild type."
Line 488: Clarify what RNA-sequence analysis is.
Line 491: Provide the reference number.
Line 491-495: This section repeats information.
In general, the last two paragraphs of the chapter 3.3.4. need to be reformatted since they both discuss the study of the same protein (DCL2) in virus resistance. Currently, they are written poorly and disjointedly. The authors need to rewrite this section into a single, logically connected text.
Comments on the Quality of English LanguageThe text still contains a large number of punctuation errors, such as missing or extra spaces, missing periods, etc.
Author Response
The revised manuscript has been significantly improved. However, a thorough review and correction of punctuation, formatting, and content structure are recommended to elevate the article to a higher standard of academic excellence.
Thanks so much for the kind feedback. The authors highly appreciate the efforts and dedicated time spent reviewing our manuscript. All the authors confirm that all required modification were done in this final version as below;
Line 128-129: It seems that the reference to Agrobacterium-mediated transformation of tomato cotyledons is missing here.
Done Line 128.
In Figure 2: The word "weed" is not capitalized, while other examples (Fungi, Bacteria, Viruses, etc.) are written with a capital letter.
Done.
Line 179: Remove the Latin name of tomato, as it is redundant here.
Done Line 179.
Line 212: Both full and abbreviated names of proteins (PRP1 and DEA1) are written without italics.
Done Line 213.
Replace the abbreviation "wt" with the full spelling "wild type."
Done Line 221.
Line 265: The word "gene" is written here without italics.
Done Line 272.
Line 270: The Latin name of the fungus is written in italics.
Done Line 277.
Line 274: The "pmr4" mutation and the Latin name of the fungus are written in italics.
Done Line 281.
References 71 and 75 are the same.
Modified.
Additionally, the last paragraph in Chapter 3.2.1 repeats the second paragraph. The chapter needs to be rewritten to avoid repetition and to cite correctly.
Modified.
Line 290-293: This is the first mention of microRNA and therefore requires a full description. In addition, "miRNA" (short for microRNA) is written with a lowercase letter. The authors' statement here is not entirely correct. Mature tomato miR482b and miR482c have a specific length of 22 nucleotides, while in general miRNA can have a length of 20 to 24 nucleotides. The beginning of this paragraph needs to be rephrased for clarity.
Modified lines 304- 316.
Line 318-319: Protein names should be written without italics.
Modified line 336.
Line 324-339: There is a repetition of sentences. Fusarium wilt has been shortened to FW; use the abbreviation consistently.
Done. Repeated sentence was deleted and the abbreviations were applied. Lines 343-355.
Line 396: "B. cinerea" should be written in italics.
Done.
Line 405: A reference to the study is missing.
Done. Line 422.
Line 412: "Tomato" should be written with a lowercase letter.
Done. Line 429.
Line 416: "Arabidopsis" should be written in italics.
Done. Line 434.
Line 417: "TOM1" should be written without brackets, and the extra period and bracket at the end of the sentence should be removed. If referring to a protein, "TOM1" should be written without italics.
Done. Line 435.
Line 424: From here on, replace "wt" with "wild type."
Done.
Line 488: Clarify what RNA-sequence analysis is.
Clarified, lines 507-508.
Line 491: Provide the reference number.
Done Line 510.
Line 491-495: This section repeats information.
Repeated information was deleted.
In general, the last two paragraphs of the chapter 3.3.4. need to be reformatted since they both discuss the study of the same protein (DCL2) in virus resistance. Currently, they are written poorly and disjointedly. The authors need to rewrite this section into a single, logically connected text.
This part was rewritten based on the suggestion.